# Optimistic Games for Combinatorial Bayesian Optimization with Application to Protein Design

**Melis Ilayda Bal** [1]*, **Pier Giuseppe Sessa**[2], **Mojmír Mutný**[3], **Andreas Krause**[3]
[1]Max Planck Institute for Intelligent Systems, Tübingen, Germany
[2]Google DeepMind    [3]ETH Zürich
`mbal@tuebingen.mpg.de, piergs@google.com`
`mmutny@inf.ethz.ch, krausea@ethz.ch`

## Abstract

Bayesian optimization (BO) is a powerful framework to optimize black-box expensive-to-evaluate functions via sequential interactions. In several important problems (e.g. drug discovery, circuit design, neural architecture search, etc.), though, such functions are defined over large *combinatorial and unstructured* spaces. This makes existing BO algorithms not feasible due to the intractable maximization of the acquisition function over these domains. To address this issue, we propose GameOpt, a novel game-theoretical approach to combinatorial BO. GameOpt establishes a cooperative game between the different optimization variables, and selects points that are game *equilibria* of an upper confidence bound acquisition function. These are stable configurations from which no variable has an incentive to deviate – analog to local optima in continuous domains. Crucially, this allows us to efficiently break down the complexity of the combinatorial domain into individual decision sets, making GameOpt scalable to large combinatorial spaces. We demonstrate the application of GameOpt to the challenging *protein design* problem and validate its performance on four real-world protein datasets. Each protein can take up to $20^X$ possible configurations, where $X$ is the length of a protein, making standard BO methods infeasible. Instead, our approach iteratively selects informative protein configurations and very quickly discovers highly active protein variants compared to other baselines.

## 1 Introduction

Many scientific and engineering problems such as drug discovery (Negoescu et al., 2011), neural architecture search (Kandasamy et al., 2018), or circuit design (Lyu et al., 2018) require optimization of expensive-to-evaluate black-box functions over combinatorial unstructured spaces involving binary, integer-valued, and categorical variables. As a concrete example, consider the *protein design* problem, *i.e.*, finding the optimal amino acid sequence to maximize the functional capacity (fitness) of the protein. Such fitness functions are highly complex, one can, in most cases, only be elucidated from real-world protein synthesis experiments. Moreover, exhaustive exploration is infeasible for both traditional lab methods and computational techniques (Romero et al., 2013) due to *combinatorial explosion*: a typical protein has 300 amino acid sites, each to be filled with one of twenty natural amino acids, yielding $20^{300}$ candidate variants.

Bayesian optimization (BO) is an established framework for optimizing black-box functions with the goal of minimizing the number of evaluations needed to certify optimality (Mockus, 1974). BO constructs a probabilistic surrogate model as a representation of the underlying black-box function, e.g., using Gaussian Processes (GPs) (Rasmussen et al., 2006). Then, it iteratively selects the next evaluations typically by maximizing a designated acquisition function. The BO framework has proven to be very powerful and successful in a variety of real-world problems including material discovery (Frazier & Wang, 2015), adaptive experimental design (Greenhill et al., 2020), or drug

---

*Corresponding author. Code is available at `https://github.com/melisilaydabal/gameopt`.

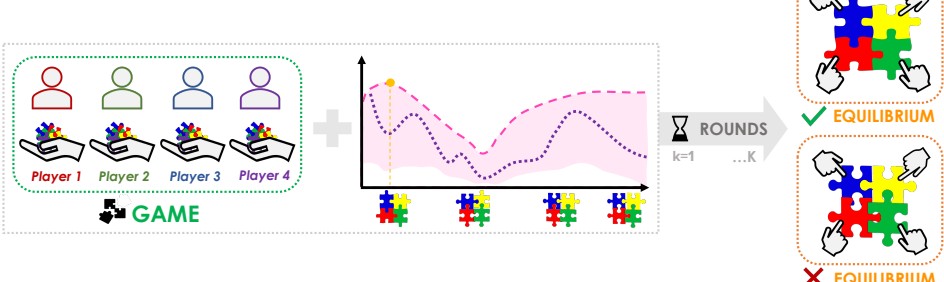

Figure 1: Illustration of GAMEOPT. GAMEOPT defines a game among the decision variables, where game rewards are represented by the upper confidence bound (UCB) function. This decouples the combinatorial decision space into individual decision sets and allows GAMEOPT to *tractably* compute game equilibria. These can be thought of as local optima of the AF in unstructured domains.

discovery (Korovina et al., 2020; Stanton et al., 2022). When considering combinatorial domains, however, standard BO methods are intractable since maximizing the acquisition function requires an exhaustive search over the whole combinatorial space (*e.g.* of size $20^{300}$ in the context of proteins) without further assumptions.

To address this challenge, we propose GAMEOPT, a novel game-theoretical framework for combinatorial BO. To circumvent the intractable maximization of an acquisition function, GAMEOPT defines a cooperative game between the discrete domain variables and, at each interaction round, selects informative points to be game *equilibria* of the acquisition function. These are stable configurations from which no player (variable) has the incentive to deviate. They can be thought of as a formalization of a notion of local optima of the acquisition function in unstructured domains (i.e., domains lacking a lattice structure). Crucially, these can be computed employing well-known equilibrium finding subroutines, effectively simulating a repeated game among the players. In this work, we utilize the Upper Confidence Bound (UCB) acquisition function, which represents an *optimistic* estimate of the underlying objective and was shown to efficiently balance exploration with exploitation (Srinivas et al., 2009). For an overview of the method, see Figure 1.

**Contributions**   We make the following main contributions:

- We propose GAMEOPT, a **novel** game-theoretical BO framework for large combinatorial and unstructured search spaces. GAMEOPT computes informative evaluation points as the equilibria (*i.e.*, local optima) of a cooperative game between the discrete variables. This overcomes the scalability issues of maximizing acquisition functions over combinatorial domains and provides a **tractable** optimization. GAMEOPT is a **flexible** procedure where the resulting per-iteration game can be solved by any readily available game strategy or solver.
- Under common kernel regularity assumptions, we bound the sample complexity of GAMEOPT, quantifying the gap between the computed equilibria (of the surrogate UCB function) and those of the underlying unknown objective. Given target accuracy level $\epsilon$, GAMEOPT returns $\epsilon$-approximate equilibria after $T = \Omega(\gamma_T \epsilon^{-2})$ iterations, where $\gamma_T$ is the kernel-dependent maximum information gain (Srinivas et al., 2009).
- We apply GAMEOPT to the challenging **protein design** problem, involving search spaces of categorical inputs. There, GAMEOPT advances the protein design process by mimicking natural evolution via a game between protein sites. We **experimentally** validate its performance on several **real-world** protein design problems based on human binding protein GB1 (Wu et al., 2016; Olson et al., 2014), iron-dependent halogenase (Büchler et al., 2022) and green-fluorescent protein (Prasher et al., 1992; Biswas et al., 2021). GAMEOPT converges consistently faster, *i.e.*, it requires fewer BO iterations to identify highly binding protein variants compared to baseline methods such as classical directed evolution.

## 2   PROBLEM STATEMENT AND BACKGROUND

**Problem statement**   We consider the problem of optimizing a costly-to-evaluate, black-box function $f : \mathcal{X} \to \mathbb{R}$ over a combinatorial unstructured space $\mathcal{X}$ without a lattice form. Suppose each element $x \in \mathcal{X}$ can be represented by $n$ discrete variables $(x^1, x^2, \ldots, x^n)$ ($n$-dimensional), where each $x^i$ takes values from a set $\mathcal{X}^{(i)}$, this makes the domain of $n \geq 1$ variables $\mathcal{X} = \mathcal{X}^{(1)} \times \ldots \mathcal{X}^{(n)}$.

---

**Algorithm 1** GAMEOPT

---

1: **Input:** GP prior $\mathcal{GP}^0(\mu_0, k(\cdot, \cdot))$, initial data $\mathcal{D}_0 = \{(x_i, y_i = f(x_i) + \epsilon)\}$, batch size $B \in \mathbb{N}$, $M \in \mathbb{N} > B$, parameter $\beta$.
2: **for** iteration $t = 1, 2, \ldots, T$ **do**
3:      Construct game with reward function $\mathrm{UCB}(\mathcal{GP}^{t-1}, \beta, \cdot) : \prod_{i=1}^n \mathcal{X}^{(i)} \to \mathbb{R}$
4:      Compute $M$ equilibria $\{x_{t,i}\}_{i=1}^M$ of the above.      */ Equilibrium-finding subroutine
5:      Select batch of top $B$ equilibria $\{x_{t,i}\}_{i=1}^B$ according to $\mathrm{UCB}(\mathcal{GP}^{t-1}, \beta, \cdot)$.      */ Filtering
6:      Obtain evaluations $y_{t,i} = f(x_{t,i}) + \epsilon_{t,i}, \quad \forall i = 1, \ldots, B$
7:      Update $\mathcal{D}_t \leftarrow \mathcal{D}_{t-1} \cup \{(x_{t,i}, y_{t,i})\}_{i=1}^B$
8:      Posterior update of model $\mathcal{GP}^t$ with $\mathcal{D}_t$.
9: **end for**

---

Assuming $|\mathcal{X}^{(i)}| = d, \ \forall i$, the size of the combinatorial space $\mathcal{X}$ is $d^n$. However, the proposed GAMEOPT framework can also operate under varying $|\mathcal{X}^{(i)}|$ sizes, as we detail in Appendix E.7.

As a concrete motivating example, consider the protein design problem (Section 5). There, $f(x)$ represents the fitness value of the designed protein sequence $x$. The search space size is $|\mathcal{X}| = 20^n$, with $n$ protein sites and $|\mathcal{X}^{(i)}| = 20$ amino acid choices per site. Moreover, a (noisy) evaluation $f(x)$ is a labor-intensive process, requiring extensive efforts and specialized laboratory equipment.

**Gaussian Processes (GPs)** Bayesian Optimization (Mockus, 1974) is a versatile framework for optimizing complex, noisy, and expensive-to-evaluate functions. BO leverages Bayesian inference to model the underlying function with a surrogate, *e.g.*, a Gaussian Process (GP) and iteratively selects evaluation points that are the most informative in terms of reducing uncertainty or enhancing model performance.

Formally, a Gaussian Process $\mathcal{GP}(\mu(\cdot), k(\cdot, \cdot))$ over domain $\mathcal{X}$ is specified by a prior mean function $\mu(x) : \mathcal{X} \to \mathbb{R}$ and a covariance function $k(x, x') : \mathcal{X} \times \mathcal{X} \to \mathbb{R}$, denoted by $f(x) \sim \mathcal{GP}(\mu(x), k(x, x'))$, where $f(x)$ represents the function value at input $x$.

Given a set of observed data points $X_t$ up to iteration $t$ and their corresponding vector of noisy observations $Y_t = f(X_t) + \epsilon_t$ with Gaussian noise $\epsilon_t \sim \mathcal{N}(0, \sigma_t^2)$, and a GP prior defined by $\mathcal{GP}(\mu_t(x), k_t(x, x'))$, the posterior distribution of the GP at iteration $t + 1$ given new observations $X_\dagger$ is again Gaussian $p(f_{t+1} \mid X_t, X_\dagger, Y_t) = \mathcal{N}(\mu_{t+1}, \sigma_{t+1}^2)$ with posterior mean and variance (Rasmussen et al., 2006).

**Bayesian Optimization (BO)** To maximize $f$, BO algorithms iteratively select evaluation points so as to balance exploration and exploitation. At each iteration the method selects the maximizer of an *acquisition function*, for example, the widely-adopted Upper-Confidence Bound (UCB) (Srinivas et al., 2009) function. Given a $\mathcal{GP}$ model at iteration $t$, the UCB function is defined as

$$\mathrm{UCB}_t(\mathcal{GP}, x) = \mu_t(x) + \beta_t \sigma_t(x), \tag{1}$$

where $\mu(x)$ and $\sigma(x)$ are the posterior mean and standard deviation at point $x$ according to $\mathcal{GP}$, and $\beta_t \in \mathbb{R}$ is a confidence parameter influencing the width of the set that can be selected to ensure the validity of the confidence set. The UCB function defines an *optimistic* estimate of the underlying objective $f$, and can effectively balance exploration (*i.e.*, favoring points with large uncertainty $\sigma_t(x)$) with exploitation (*i.e.*, selecting points with large posterior mean $\mu_t(x)$).

While standard BO methods can efficiently optimize $\mathrm{UCB}(\mathcal{GP}, \cdot)$ in efficiently enumerable or continuous domains, they become very soon intractable in the case of combinatorial unstructured domains, such as the space of possible amino acid sequences. In the next section, we propose GAMEOPT, a novel BO approach that circumvents such prohibitive difficulty.

## 3 GAMEOPT ALGORITHM

In a nutshell, the proposed GAMEOPT (Optimistic Games) approach circumvents the combinatorial optimization of the UCB function by defining a *cooperative game* among the $n$ input variables and computes the associated equilibria as candidate evaluation points. Formally, at each iteration $t$, GAMEOPT defines a cooperative game (Fudenberg & Tirole, 1991) involving $\mathcal{N} = \{1, \ldots, n\}$

players, each player $i$ taking actions in the discrete set $\mathcal{X}^{(i)}$. In such a game, the players' interests are aligned towards the goal of maximizing the function $\text{UCB}(\mathcal{GP}^t, \cdot) : \prod_{i=1}^n \mathcal{X}^{(i)} \to \mathbb{R}$, where $\mathcal{GP}^t$ is the current GP estimate at iteration $t$. Thus, it can be interpreted as an optimistic game with respect to the true unknown $f$. In such a game, the goal of the players is to compute game (Nash) *equilibria*, defined as follows.

**Definition 3.1** (Nash equilibrium (Nash, 1951))**.** Let $r^i : \mathcal{X} \to \mathbb{R}$ be the reward function of each player $i$, defined over joint configuration $x$. A joint strategy profile $x_{\text{eq}} = (x_{\text{eq}}^1, \ldots, x_{\text{eq}}^n)$ is a Nash equilibrium if, for every player $i \in \mathcal{N}$, $r^i(x_{\text{eq}}^i, x_{\text{eq}}^{-i}) \geq r^i(x^i, x_{\text{eq}}^{-i}), \forall x^i \in \mathcal{X}^{(i)}$, where $x_{\text{eq}} = (x_{\text{eq}}^i, x_{\text{eq}}^{-i})$, and $x_{\text{eq}}^{-i}$ is the joint equilibrium strategy of all players except $i$.

The existence of such equilibrium point(s) is guaranteed since players and actions are finite (Fudenberg & Tirole, 1991). Moreover, because players' reward functions are aligned and coincide with $\text{UCB}(\mathcal{GP}^t, \cdot)$, efficient polynomial-time equilibrium-finding methods can be employed, such as Iterative Best-Response (IBR), where players update their actions sequentially, or simultaneous multiplicative weights updates such as the HEDGE (Freund & Schapire, 1997) algorithm. We report these two possible strategies in Algorithms 2 and 3 in Section 3.1. Intuitively, equilibria are computed by breaking down the complex decision space into individual decision sets, as illustrated in Figure 1. Mathematically, we refer to this operation as $\arg\text{eq}$, returns the joint configuration(s) which form equilibria according to UCB,

$$x_{\text{eq}} = \arg\text{eq}_{x^i \in \mathcal{X}^{(i)}; i \in \mathcal{N}} \text{UCB}(\mathcal{GP}^t, (x^1, \ldots, x^n)). \tag{2}$$

Our overall approach is summarized in Algorithm 1. In practice, we compute $M > 1$ equilibria and subselect a batch of top $B < M$ equilibria according to the $\text{UCB}(\mathcal{GP}^t, \cdot)$ criterion. Subsequently, such a batch is evaluated by $f$, the GP model is updated accordingly, and a new game with an updated reward function is defined at the next iteration based on the updated posterior.

**A form of local optimality** Within GAMEOPT, each player strategically selects actions to maximize their collective payoff, much like seeking local optima in a continuous multi-dimensional function (see Figure 1). In continuous optimization, a local optimum is a point, where there is no direction that leads to an improvement, similarly, as in our framework there is not a player that can unilaterally improve the value of the collective pay-off. In essence, seeking equilibria is analogous to seeking local optima of a continuous acquisition function, and our game-based approach allows us to effectively pinpoint them within an unstructured combinatorial space. We remark that GAMEOPT computes equilibria of the current $\text{UCB}(\mathcal{GP}^t, \cdot)$ function which, as we show in Section 5, are better and better approximations of equilibria of the unknown objective $f$.

**Price of Anarchy** But how good are equilibria compared to the global optimum? The quality of equilibria (also known as the efficiency of the game) can be quantified via the game-theoretic notion of Price of Anarchy (PoA) (Christodoulou & Koutsoupias, 2005), defined as the ratio between the worst equilibrium and the global optimum, *i.e.*, $\text{PoA} := \min_{\mathbf{x} \in \mathcal{E}} f(\mathbf{x}) / \max_{\mathbf{x}} f(\mathbf{x})$ where $\mathcal{E}$ is the set of all equilibria of $f$. PoA has been extensively studied for various classes of games and can sometimes be upper-bounded given further assumptions on $f$. As an example, in case $f$ is a submodular function (over binary, integer, or continuous domains), PoA is guaranteed to be at least 0.5 (Vetta, 2002; Sessa et al., 2019b). Although such a PoA guarantee does not readily apply to our setting, we believe similar ones could be proved for the case of unstructured domains — though this is beyond the scope of our work. In practice, given an unknown function $f$ (such as the protein's fitness function in our experiments of Section 5), not all equilibria may achieve high function values (i.e. PoA can be very low). Nevertheless, GAMEOPT computes *multiple* equilibria ($M > 1$) at each iteration and selects only the top $B$ according to their UCB value. We believe this is a key form of robustness that can effectively filter out suboptimal equilibria and empower GAMEOPT's experimental performance.

### 3.1 EQUILIBRIUM FINDING SUBROUTINES

We present a set of established algorithms for finding an equilibrium of the game introduced in Eq. (2).

**Iterative best responses** One possible subroutine for Algorithm 1 is Iterative Best Response (IBR) procedure as provided in Algorithm 2. Concretely, under the cooperative game setting outlined in Section 3 and given $\mathcal{GP}$-predicted UCB function, each player *iteratively* selects the response that maximizes the value of the game given that the other players play the joint strategy from the previous

---

**Algorithm 2** Iterative Best Response (IBR)

1: **Input:** Domain $\mathcal{X}$, payoff (reward) $r : \mathcal{X} \to \mathbb{R}$, players $\mathcal{N}$.
2: $\mathbf{x}_0^{br} \leftarrow$ random joint strategy, $\mathbf{x}_0^{br} \in \mathcal{X}$.
3: **for** round $k = 1, \ldots, K$ **do** */ BR game
4:     $\mathcal{X}_k^{br} \leftarrow \Big\{ (x^{i,br}, x_{k-1}^{-i,br}), \text{ such that}$
    $x^{i,br} = \arg \max_{x \in \mathcal{X}^{(i)}} r(x, x_{k-1}^{-i,br}) \Big\}_{i=1}^n$
5:     Play $\mathbf{x}_k^{br} \leftarrow \arg \max_{x_k \in \mathcal{X}_k^{br}} [r(x_k)]$
6: **end for**
7: **return** $\mathbf{x}_K^{br}$       */ Equilibrium

---

**Algorithm 3** Simultaneous HEDGE

1: **Input:** Domain $\mathcal{X} = \prod_{i=1}^n \mathcal{X}^{(i)}$ with $| \mathcal{X}^{(i)} | = W$, payoff $r : \mathcal{X} \to \mathbb{R}$, players $\mathcal{N}$, parameter $\eta$.
2: Initialize weights $\boldsymbol{w}_1 \leftarrow \frac{1}{W}[1, \ldots, 1] \in \mathbb{R}^{|\mathcal{N}| \times W}$
3: **for** round $k = 1, \ldots K$ **do**     */ Compute CCE
4:     Sample $x_k^i \sim \boldsymbol{w}_1^i, \forall i \in \mathcal{N}$
5:     Set joint strategy $\mathbf{x}_k \leftarrow \{x_k^i\}_{i \in \mathcal{N}}$
6:     **for** player $i \in \mathcal{N}$ **do**     */ Players' payoff
7:       $\ell_{x_k^{-i}} \leftarrow [r(x_k^{j,-i})]_{\forall j \in \mathcal{X}^{(i)}}$, where
      $x_k^{j,-i} = j \cup \{x_k^{i'}\}_{i' \in \mathcal{N} \setminus \{i\}}, \forall j \in \mathcal{X}^{(i)}$
8:       Set $\boldsymbol{w}_{k+1}^i \propto \boldsymbol{w}_k^i \exp(\eta \ell_{x_k^{-i}})$
9:     **end for**
10: **end for**
11: **return** Uniform$\{\mathbf{x}_1, \ldots, \mathbf{x}_K\}$     */ Equilibrium

---

round. Each player is sequentially selected to play their best response in a round-robin fashion. Because action space is finite, this procedure is guaranteed to converge to a local maximum of the UCB function *i.e.*, an equilibrium of the underlying game (Fudenberg & Tirole, 1991).

**Multiplicative weights updates** Alternatively, we can compute game equilibria letting players *simultaneously* act according to a multiplicative weights update algorithm such as HEDGE (Freund & Schapire, 1997), see Algorithm 3. We can cast equilibrium computation as an instance of *adversarial online learning* among multiple learners (Cesa-Bianchi & Lugosi, 2006). Here, each player selects a strategy based on their available options and, after observing the joint payoff, players' strategies are re-weighted based on past performance. Through repeated rounds of play and re-weighting, the empirical frequency of play forms a coarse correlated equilibrium (CCE) (a weaker notion of Nash equilibrium), see e.g. (Cesa-Bianchi & Lugosi, 2006), while convergence to pure Nash equilibria is also guaranteed in some cases (Kleinberg et al., 2009; Palaiopanos et al., 2017).

### 3.2 RELATED WORK

While there exist rather few works in the area (Papenmeier et al., 2023), existing combinatorial BO methods either target surrogate modeling with discrete variables (Baptista & Poloczek, 2018; Oh et al., 2019; Garrido-Merchán & Hernández-Lobato, 2020; Kim et al., 2021; Deshwal et al., 2023) or optimizing acquisition function within discrete spaces (Baptista & Poloczek, 2018; Deshwal et al., 2020; 2021a;b; Khan et al., 2023). However, they often require a parametric surrogate model with higher-order interaction specifications for combinatorial structures (Baptista & Poloczek, 2018) or domain-specific knowledge (Deshwal et al., 2020). In contrast, GAMEOPT relies on a non-parametric surrogate model, without the need for domain-specific knowledge.

Closest to ours is (Daulton et al., 2022), which also targets optimizing the acquisition function in high-cardinality discrete/mixed search spaces via a probabilistic reparameterization (PR) that maximizes the expectation of the acquisition function. However, PR fails at being tractable since it requires evaluating the expectation over the joint distribution of all decision variables, requiring combinatorially many elements to be summed. An accurate estimate would require extensive sampling without special structural assumptions. In contrast, GAMEOPT treats each variable *independently* (potentially in parallel) within the game, keeping the values of the remaining variables fixed during each strategy update. We use PR as a baseline to evaluate our approach in Section 5, and demonstrate improved performance of our method on protein design problems.

Further, a body of research has focused on BO over continuous (latent) spaces (Gómez-Bombarelli et al., 2018; Eriksson et al., 2019; Tripp et al., 2020; Deshwal & Doppa, 2021; Maus et al., 2022; Stanton et al., 2022; Zhang et al., 2023). These methods learn continuous sequence embeddings and optimize with gradient-based techniques by utilizing deep generative models. However, the primary problem we address in our study is the intractable acquisition function optimization over *large combinatorial* search spaces, specifically tackling the challenge of exhaustive exploration. In line with this, we select our baselines accordingly and include a comparison with some latent space optimizers only as additional experimental evaluation for insight.

Recently, a line of works has explored the interplay between BO & game theory (Sessa et al., 2019a; Dadkhahi et al., 2020; Han et al., 2024), and coordinated exploration (Sessa et al., 2022; Bal et al., 2024) for various domains, but their connection with combinatorial BO is novel.

## 4    SAMPLE-COMPLEXITY GUARANTEES

In this section, we derive sample-complexity guarantees for GAMEOPT. Namely, we characterize the number of interaction rounds $T$ required to reach approximate equilibria (*i.e.*, local optima) of the true function $f$. For simplicity, we assume GAMEOPT is run with batch size $B = 1$, though our results can be generalized to larger $B$.

The obtained guarantees are based on standard regret bounds of Bayesian optimization adapted to our equilibrium finding goal. These are characterized by the widely utilized notion of maximum information gain (Srinivas et al., 2009):

$$\gamma_t = \frac{1}{2} \log \left| I_t + \sigma^{-1} \boldsymbol{K}_t \right|. \tag{3}$$

This is a kernel-dependent ($\boldsymbol{K}_t$) quantity that quantifies the maximal uncertainty reduction about $f$ after $t$ observations. Further, to characterize our sample complexity, we define the notion of $\epsilon$-approximate Nash equilibrium.

**Definition 4.1** ($\epsilon$-approximate Nash equilibrium). A strategy profile $\tilde{x}_{\text{eq}}$ is a $\epsilon$-approximate (Nash) equilibrium of $f$ if, for each $i \in \mathcal{N}$, $f(\tilde{x}_{\text{eq}}) \geq f(x^i, \tilde{x}_{\text{eq}}^{-i}) - \epsilon, \forall x^i \in \mathcal{X}^{(i)}$.

In the next main theorem, we provide a lower bound on the number of iterations $T$ to reach approximate equilibria. After $T$ rounds, we assume GAMEOPT returns $x_{T^\star}$ with:

$$T^\star := \arg \min_{t \in [T]} \max_{i, x^i} \left[ \text{UCB}(\mathcal{GP}^t, x^i, x_t^{-i}) - \text{LCB}(\mathcal{GP}^t, x_t) \right],$$

where LCB is the lower confidence bound function $\text{LCB}(\mathcal{GP}^t, x) = \mu_t(x) - \beta_t \sigma_t(x)$. That is, among the selected points $x_1, \ldots, x_T$, $x_{T^\star}$ is the one that guarantees the minimum worst-case single-player deviation. The deviation above is computed according to the UCB and with respect to the LCB, thus representing an upper bound on the actual deviation in terms of $f$. We can affirm the following.

**Theorem 4.2** (Sample complexity of GAMEOPT). *Assume $f$ satisfies the regularity assumptions of Section 2, and GAMEOPT is run with confidence width $\beta_t = 2n \log \left( \sup_{i \in \mathcal{N}} |\mathcal{X}_i| \frac{t^2 \pi^2}{6\delta} \right)$. Then, with probability at least $1 - \delta$ and for a given accuracy $\epsilon \geq 0$, the strategy $x_{T^\star}$ returned by GAMEOPT is a $\epsilon$-approximate Nash equilibrium when*

$$T \geq \Omega \left( \frac{\beta_T \gamma_T}{\epsilon^2} \right). \tag{4}$$

An equivalent interpretation of the above result is as follows: After $T$ iterations, GAMEOPT returns an $\epsilon_T$-approximate Nash equilibrium of $f$, with approximation factor $\epsilon_T \leq \mathcal{O}(T^{-\frac{1}{2}} \sqrt{\beta_T \gamma_T})$. Note that the latter bound is the typical rate of convergence of BO algorithms (Srinivas et al., 2009) to the global maximizer. Instead, in our combinatorial BO setup –where global optimization is intractable– it corresponds to the rate of convergence to equilibria. A more explicit convergence guarantee can be obtained by employing existing bounds for $\gamma_T$ which are known for commonly used kernels (Srinivas et al., 2009). E.g., for squared exponential kernels $\gamma_T = \mathcal{O}(\log(T)^{nd})$ where $d$ is the dimension of each input space $\mathcal{X}^{(i)}$ for each player $i \in \mathcal{N}$, with $|\mathcal{N}| = n$.

## 5    APPLICATION TO PROTEIN DESIGN

In this section, we specialize the GAMEOPT framework to protein design, a problem defined over the space of possible amino acid sequences. Note that such domains are highly combinatorial (their size grows exponentially with the sequence length) and unstructured (i.e. they lack a lattice structure). In this context, computing game equilibria follows the natural principle of promoting beneficial mutants and mirrors the proteins' mutation and selection process. In Algorithms 4 and 5 (Appendix B), we provide a detailed elaboration of GAMEOPT for protein design using equilibrium-finding methods. We showcase its performance in four real-world protein datasets.

In the protein design context, GAMEOPT establishes a cooperative game among the different protein sites $i \in \{1, \ldots, n\}$, where $n$ is the length of the protein sequence. Each site $i$ chooses an amino acid from the set with $| \mathcal{X}^{(i)} |= 20$, where the switching between amino acid choices can be thought of as biological *mutation*. The joint objective of the players is to converge to a highly rewarding protein sequence, as measured by the GP-predicted optimistic score for the fitness function. This mirrors the *selection* phase in *evolutionary search*, providing a *directed* approach to protein optimization. Below, we discuss related work in the area.

**Related work on evolutionary search.** A considerable line of works (Arnold, 1998; Hansen, 2006; Romero & Arnold, 2009; Yang et al., 2019; Deshwal et al., 2020; Cheng et al., 2022; Low et al., 2023) centers around evolutionary search algorithms for optimizing black-box functions. Within combinatorial amino-acid sequence spaces, the highly regarded technique, *directed evolution* (Arnold, 1998; Romero & Arnold, 2009), draws inspiration from natural evolution and identifies local optima through a series of repeated random searches, characterized by controlled iterative cycles of mutation and selection. Expanding upon this, machine learning-guided variants (Yang et al., 2019; Wittmann et al., 2021; Romero et al., 2013; Angermueller et al., 2020) mitigate the sample-inefficiency and intractability concerns associated with directed evolution. In general, these methods are not data-driven in the sense that they do not use the whole extent of the past data and focus on the best variant found so far or a selection thereof and propose a random search from thereon. Alternatively, even if allowed to adapt to the past outcomes, they tend to restrict themselves to very small search spaces (Büchler et al., 2022). Instead, our approach uses all past data to create a UCB estimate of the fitness landscape and utilize it to simulate a cooperative evolution in problems where even the whole sequence of the protein can be optimized. Moreover, compared to such methods, GAMEOPT mimics evolution *within each interaction* using the surrogate UCB function.

## 5.1 DATASETS

We empirically evaluate GAMEOPT on real-world protein design problems, specifically focusing on the following instances: protein G domain B1, *GB1*, binding affinity to an antibody IgG-FC (KA), examined on two distinct datasets, *GB1(4)* (Wu et al., 2016) and *GB1(55)* (Olson et al., 2014), characterized by sequence lengths of 4 and 55, respectively; three critical amino acid positions in an iron/$\alpha$-ketoglutarate-dependent halogenase with sequence length 3 (Büchler et al., 2022); and *Aequorea victoria* green-fluorescent protein (*GFP*) of length 238 (Prasher et al., 1992; Biswas et al., 2021). The former *GB1* dataset is fully combinatorial, *i.e.*, covering fitness measurements of $20^4$ variants. Here, each protein site is treated as a player in the GAMEOPT. The latter is non-exhaustive, including only 2-point mutations of *GB1*. Thus, an MLP having $R^2 = 0.93$ on a test set is trained and treated as the ground truth fitness for the fully combinatorial dataset. For *GB1(55)*, we also consider a modified setup where "only" 10 sites can be mutated. Similarly, *Halogenase* and *GFP* are also non-exhaustive involving fitness measurements for 605 and 35,584 unique variants, respectively. To obtain the complete protein fitness landscape, we once again construct oracles for these datasets, utilizing MLPs achieving $R^2 = 0.96$ and $R^2 = 0.90$ on their respective test sets. In the case of the *Halogenase* dataset, each protein site is treated as a player, while for the *GFP* dataset, 6 and 8 sites are designated as players. Further experimental details are in Appendix D.

## 5.2 EXPERIMENTAL SETUP

In all experiments, we use a GP surrogate with an RBF kernel for GP-based methods. The RBF specifies lengthscales for each input variable separately – sometimes known as ARD kernels (Rasmussen et al., 2006). To handle categorical inputs to the GP surrogate, we employ feature embeddings as representations for these inputs using the ESM-1v transformer protein language model by (Meier et al., 2021). The prior mean for the GP is pre-defined as the average log fitness value over the whole dataset. Kernel hyperparameters are optimized prior to the start of optimization and remain fixed throughout the BO iterations; specifically, lengthscales are optimized over the training set at the start of each replication using Bayesian evidence, and the outputscale is fixed to the difference between the maximum fitness value observed in the dataset & mean. In other words, we also fit a prior mean. A consistent observation noise of 0.0004 is maintained for each training example. Moreover, we use batch size $B = 5$. In Appendix D, we provide the (hyper)parameter settings (see Table 1) and the detailed setup for the experiments.

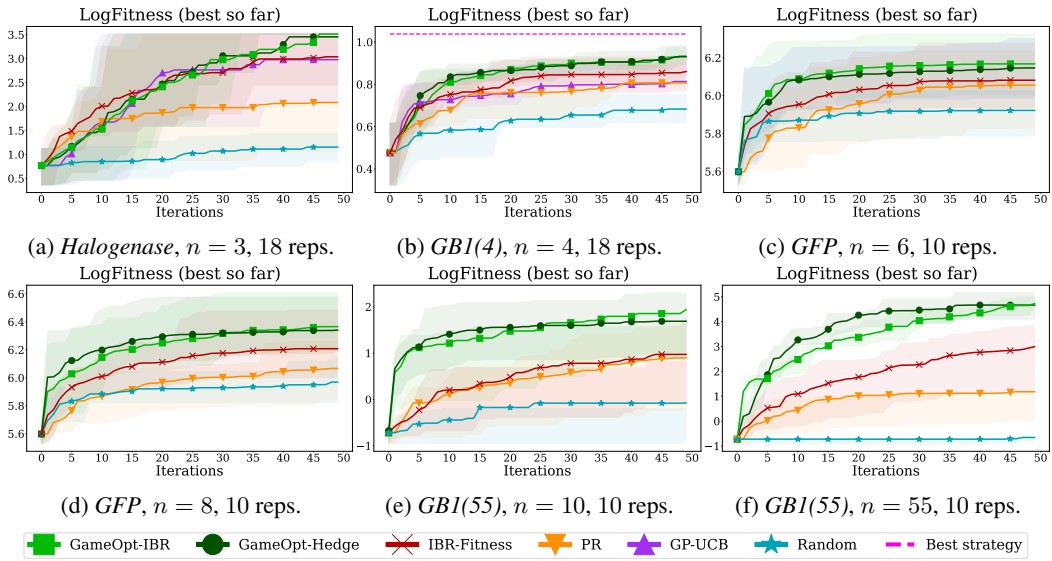

Figure 2: Convergence speed of methods in terms of log fitness value of the best-so-far protein across BO iterations, under batch size $B = 5$. Results are averaged over replications initiated with different training sets: 100 protein variants for *Halogenase* and *GB1(4)*, and 1000 protein variants for other domains. Error bars are interquartile ranges averaged over replications. In all experiments GAMEOPT with IBR and HEDGE subroutines discover better protein sequences at a much faster rate.

## 5.3 BASELINES

We benchmark GAMEOPT against the following baselines:

1. GP-UCB (Srinivas et al., 2009) selecting –at each iteration– the best $B$ points in terms of UCB value. Note that this is feasible (though computationally expensive) only for the *GB1(4)* and *Halogenase* datasets, while it is prohibitive for *GB(55)* and *GFP*
2. IBR-FITNESS, which mimics directed evolution (Arnold, 1998) through a series of local searches on the fitness landscape, iteratively selecting the $B$ best-responses based on log fitness criterion
3. PR (Daulton et al., 2022), a state-of-the-art discrete/mixed BO approach picking $B$ points using the expected UCB criterion
4. RANDOM baseline randomly sampling $B$ random sequences at each iteration.

Further details and pseudo codes of such baselines are in Appendix C.

We assess our method using two key metrics: convergence speed and sampled batch diversity w.r.t. past, (*i.e.*, the degree of distinctiveness among newly acquired samples in comparison to the original data point particularly in the context of the input space) for BO evaluation. The latter can also be regarded as the measure of exploration. Convergence speed is tracked by the log fitness value of the best-so-far discovered protein variant across BO iterations. We monitor the diversity of the sampled batch concerning the past across BO iterations through (1) the average Hamming distance between the executed variant and the proposed variant from the previous iteration (pairwise distance) and (2) the average Hamming distance of the executed variant from the nearest initial training point.

In Appendix E, we provide additional performance metrics such as the fraction of global optima discovered, the fraction of discovered solutions above a fitness threshold, cumulative maximum, and mean pairwise Hamming distances. Moreover, we compare with discrete local search methods (Balandat et al., 2020) in Appendix E.3 and report their respective runtimes in Appendix E.4.

## 5.4 RESULTS

GAMEOPT, with IBR and HEDGE equilibrium computation subroutines, consistently outperform baselines across all experiments, discovering higher fitness protein sequences faster (see Figure 2).

**Results for *Halogenase*.** While initially surpassed by IBR-FITNESS, PR, and GP-UCB, GAMEOPT variants converge faster to higher log fitness proteins than baselines. Notably, IBR-FITNESS performs

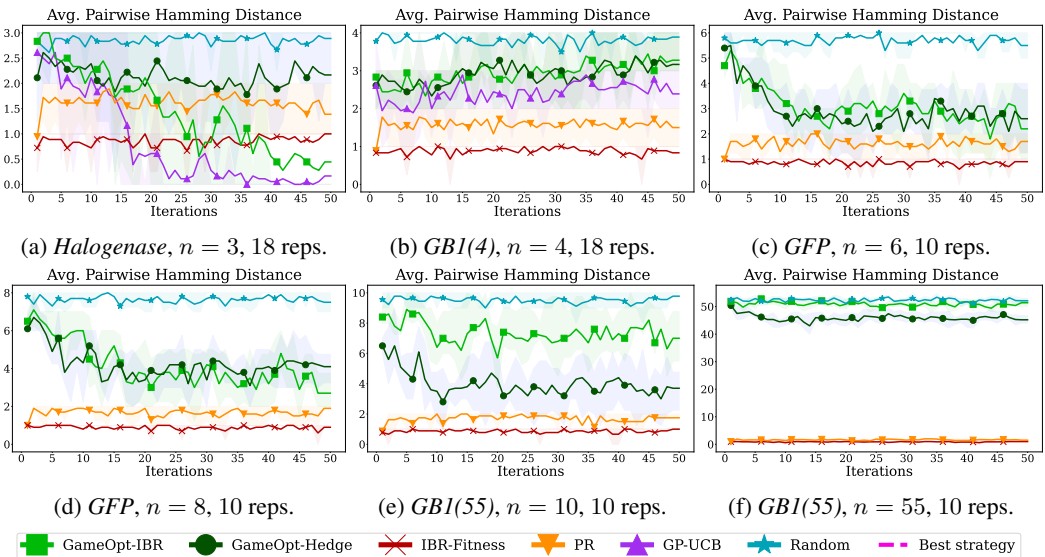

Figure 3: Sampled batch diversity relative to past, measured via mean Hamming distance between executed and proposed variants from the previous iteration (pairwise distance), under batch size $B = 5$. Results are averaged over replications, and error bars show interquartile ranges. In all experiments GAMEOPT consistently samples a rather diverse batch of evaluation points w.r.t. past proposed variants. This enhanced exploration of the search space contributes to its strong performance compared to the baseline methods.

best-responses on the true log fitness function, whereas GAMEOPT-IBR simulates best-response dynamics directly on the UCB model, allowing to compute equilibria at each iteration. Additionally, although GP-UCB performs comparably, it incurs higher computational demands. Furthermore, PR and RANDOM perform poorly in the *Halogenase* setting.

**Results for *GB1(4)*.** Similarly, GAMEOPT approaches steadily surpass PR and GP-UCB while exploring superior protein sequences efficiently. Initially trailing IBR-FITNESS, GAMEOPT approaches prove more adept at exploring and sampling diverse points (see Figure 4 in Appendix E).

The baseline PR is not very competitive and also comes with higher computational demands. As highlighted in Section 3.2, PR relies on the expected UCB as the acquisition function, requiring expectation computation across players set and amino acid choices. This makes its performance contingent on accurately estimating expected UCB through combinatorially many sequence samples. In contrast, GAMEOPT efficiently finds stable outcomes by breaking down the combinatorial search space into individual decision sets, resulting in a more manageable process.

Finally, of particular observation is the subpar performance of GP-UCB and RANDOM. A detailed analysis of GP-UCB's performance is presented in Appendix E.2, where it is observed that the efficacy of GP-UCB heavily relies on the quality of the initial GP surrogate. In contrast, GAMEOPT demonstrates robustness in overcoming the limitations of a model initialized with a limited amount of data, thereby enhancing its sample efficiency. Further discussion on the performance of methods can be found in Appendix E.

**Results for *GFP*.** The complexity of the problem positively correlates with the performance gap between GAMEOPT and baselines. In the protein search space with 6 and 8 amino acid decisions, GAMEOPT with either subroutine excels in identifying high-log fitness protein sequences even from the start. Figure 3 further demonstrates GAMEOPT's consistent exploration of diverse batches and Figure 4 in Appendix E.1 shows its high rate of exploration.

**Results for *GB1(55)*.** In both versions of the most complex problem domain, GAMEOPT demonstrates superior performance. As the decision flexibility (*i.e.*, the number of players) increases from 10 to 55, the performance gap against baselines widens. Furthermore, GAMEOPT achieves incomparable batch diversity concerning past, both with respect to the initial training and previously executed protein sequences, as shown in Figures 3 and 4 in Appendix E.1.

## 5.5 FURTHER DISCUSSION AND LIMITATIONS

In our experiments, we compare to IBR-FITNESS, which simulates currently employed strategies in the iterative protein optimization literature. This is by no means the only methodology applied in this field, and a comprehensive comparison is beyond the scope of this work.

**Batch size.**  Similar to the previous point, in the experiments presented in Section 5.4, we use a batch size of $B = 5$. While we acknowledge this is a restrictive setup given the technological needs of common screenings, nevertheless, our results should be transferable and applicable irrespective of the batch size which differs to each laboratory setting. To support this, we include batch size ablations, $B = \{1, 3, 5, 10, 50\}$, in Appendix E.8 (Figure 12), demonstrating that GAMEOPT variations consistently outperform baseline methods in discovering better protein sequences across various batch sizes.

**Learning rate.**  In Appendix E.9, we detail learning rate ablations for GAMEOPT-HEDGE.

**Efficient vs better optimization of the acquisition function.**  The primary benefit of GAMEOPT is its ability to efficiently optimize the acquisition function by adopting a game-theoretical perspective. Unlike GP-UCB, our focus is on *tractable* optimization in large combinatorial spaces rather than improving optimization. This efficiency is vital for navigating such spaces, where GAMEOPT exploits game equilibria to enhance exploration. Appendix E.10 (see Figure 15) provides an analysis of UCB values over BO iterations. While GAMEOPT is initially outperformed by GP-UCB in collecting higher UCB-valued batches, its exploration strategy ultimately leads to more efficient optimization.

**Different acquisition functions.**  We design the GAMEOPT framework using UCB as the acquisition function (also the game reward function), with its sample complexity derived accordingly. To assess broader applicability, we also evaluate GAMEOPT with alternative acquisition functions, such as expected improvement (Jones et al., 1998). As shown in Appendix E.11, Figure 16, our empirical comparison with GP-based methods reveals that GAMEOPT consistently outperforms these baselines across all performance metrics.

**Comparison with latent space optimizers.**  In line with the primary focus of this study, we compare GAMEOPT to baselines that employ acquisition function optimizers operating *directly* on large *combinatorial* search spaces (Dreczkowski et al., 2024), addressing the challenge of exhaustive exploration. As discussed in Section 3.2, another line of work follows BO over continuous spaces by learning a latent representation via deep generative models. While a comprehensive comparison with these methods is beyond our scope, we provide additional insights in Appendix E.12, where we empirically compare GAMEOPT against Naïve LSBO-(L-BFGS-B) (Gómez-Bombarelli et al., 2018), using a second-order gradient-based optimizer, and LADDER (Deshwal & Doppa, 2021). The results highlight GAMEOPT's effectiveness— not only in finding higher-fitness sequences tractably but also in bypassing the limitations of latent space optimizers, particularly their dependency on a decoder.

**Sequence-based kernels.**  To show the applicability of GAMEOPT under various kernel choices, we provide additional analysis using string kernels (Moss et al., 2020) in Appendix E.12. Specifically, we consider a structure-coupled kernel designed by combining an RBF kernel with a sub-sequence string kernel (Moss et al., 2020). In this setting, GAMEOPT demonstrates faster convergence to higher log fitness protein sequences, further emphasizing its adaptability across different kernel configurations.

## 6 CONCLUSIONS

We introduced GAMEOPT, a novel *tractable* game-theoretical approach to combinatorial BO that leverages game equilibria of a cooperative game between discrete inputs of a costly-to-evaluate black-box function to tractably optimize the acquisition function over combinatorial and unstructured spaces, and select informative points. Empirical analysis on challenging protein design problems showed that GAMEOPT surpassed baselines in terms of convergence speed, consistently identifying better protein variants more quickly, thereby being more resource-efficient. GAMEOPT is a versatile framework, allowing for exploration with different acquisition functions or mixed equilibrium concepts. As for future work, an adaptive grouping of players could be further investigated.

ACKNOWLEDGEMENTS

MIB acknowledges the support of the Max Planck Graduate Center for Computer and Information Science. The preliminary experiments were made possible through the support of the Max Planck Institute for Software Systems compute cluster.

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

# Appendix

## Table of Contents

We gather here the technical proofs, the details on GAMEOPT's application to protein design, and additional experiment results complementing the main paper.

## A  PROOF OF THEOREM 4.2

The proof relies on the main confidence lemma (Srinivas et al., 2009, Lemma 5.1) which states that, when the confidence width is set as $\beta_t = 2n \log \left( \sup_{i \in [n]} |\mathcal{X}_i| \frac{t^2 \pi^2}{6\delta} \right) \geq 2 \log \left( | \prod_{i=1}^{n} \mathcal{X}_i | \frac{t^2 \pi^2}{6\delta} \right)$, then with probability at least $(1 - \delta)$,

$$\mu_t(x) - \beta_t \sigma_t(x) \leq f(x) \leq \mu_t(x) + \beta_t \sigma_t(x), \quad \forall x \in \mathcal{X}, \quad \forall t \geq 1. \tag{5}$$

In other words, $\mathrm{UCB}(\mathcal{GP}^t, \cdot)$ and $\mathrm{LCB}(\mathcal{GP}^t, \cdot)$ are upper and lower bound functions with high probability. For simplicity, we will use the notation $\mathrm{UCB}_t(\cdot)$ and $\mathrm{LCB}_t(\cdot)$ for $\mathrm{UCB}(\mathcal{GP}^t, \cdot)$ and $\mathrm{LCB}(\mathcal{GP}^t, \cdot)$, respectively.

Next, we show that after $T$ iterations the strategy reported by GAMEOPT: $x_{T^\star}$, with $T^\star = \arg\min_{t \in [T]} \max_{i,x^i} \mathrm{UCB}_t(x^i, x_t^{-i}) - \mathrm{LCB}_t(x_t)$, is a $\epsilon_T$-approximate Nash equilibrium of $f$ with $\epsilon_T \leq \mathcal{O}(T^{-0.5}\sqrt{\gamma_T})$. The theorem statement then follows by a simple inversion of the aforementioned bound.

By definition, $x_{T^\star}$ is a $\epsilon_T$-approximate Nash equilibrium of $f$ when $\epsilon_T = \max_{i,x^i} f(x^i, x_{T^\star}^{-i}) - f(x_{T^\star})$, i.e. $\epsilon_T$ upper bounds all possible single-player deviations. We can bound the worst-case single-player deviation with probability $(1 - \delta)$ by the following chain of inequalities:

$$\epsilon_T = \max_{i,x^i} f(x^i, x_{T^\star}^{-i}) - f(x_{T^\star}) \leq \max_{i,x^i} \mathrm{UCB}_{T^\star}(x^i, x_{T^\star}^{-i}) - \mathrm{LCB}_{T^\star}(x_{T^\star}) \tag{6}$$

$$\leq \frac{1}{T} \sum_{t=1}^{T} \max_{i,x^i} \mathrm{UCB}_t(x^i, x_t^{-i}) - \mathrm{LCB}_t(x_t) \tag{7}$$

$$= \frac{1}{T} \sum_{t=1}^{T} \max_{i,x^i} \mathrm{UCB}_t(x^i, x_t^{-i}) - \mathrm{UCB}_t(x_t) + \frac{2}{T} \sum_{t=1}^{T} \beta_t \sigma_t(x_t) \tag{8}$$

$$\leq \frac{2}{T} \sum_{t=1}^{T} \beta_t \sigma_t(x_t) \leq \mathcal{O}(T^{-0.5}\sqrt{\beta_T \gamma_T}). \tag{9}$$

The first inequality follows from the confidence lemma (5). The second one, by the fact that $\max_{i,x^i} \mathrm{UCB}_{T^\star}(x^i, x_{T^\star}^{-i}) - \mathrm{LCB}_{T^\star}(x_{T^\star}) \leq \max_{i,x^i} \mathrm{UCB}_t(x^i, x_t^{-i}) - \mathrm{LCB}_t(x_t), \forall t$, by definition of $T^\star$. The last inequality holds because, at each iteration $t$, $x_t$ is an equilibrium of the $\mathrm{UCB}_t$ function. Finally, the last inequality is from (Srinivas et al., 2009, Lemma 5.4). □

## B  GAMEOPT FOR PROTEIN DESIGN

The core concept of the GAMEOPT framework is inspired by the principles of natural evolution. In protein design, achieving equilibrium of a cooperative game over protein sites mirrors the iterative mutation and selection process in evolution. Where it converges to beneficial mutant sequences, can be thought of as equilibrium of the game. Given that protein search spaces align well with the domain GAMEOPT works on, we introduce a specialized version of GAMEOPT, tailored for protein design applications.

## C  BASELINES

In Section 5, we empirically evaluate GAMEOPT against existing baselines which we detail next. These include IBR-FITNESS, inspired by directed evolution (Algorithm 6), RANDOM (Algorithm 7), which samples evaluation points randomly, and PR, an optimizer of expected UCB (Daulton et al., 2022). We further compared GAMEOPT with discrete local search methods in Appendix E.3.

---

**Algorithm 4** GAMEOPT-IBR for Protein Design

---

1: **Input:** GP prior $\mathcal{GP}^0(\mu_0, k(\cdot, \cdot))$, initial data $\mathcal{D}_0 = \{(x_i, y_i = f(x_i) + \epsilon)\}$, protein sites $\mathcal{N}$, batch size $B \in \mathbb{N}$, $M \in \mathbb{N} > B$, parameter $\beta$.
2: **for** iteration $t = 1, 2, \ldots, T$ **do**
3:  Construct game with reward function $\text{UCB}(\mathcal{GP}^{t-1}, \beta, \cdot) : \prod_{i=1}^n \mathcal{X}^{(i)} \to \mathbb{R}$
4:  **for** $m = 1, 2, \ldots, M$ **do**
5:   $\mathbf{x}_0^{br} \leftarrow$ random starting protein sequence, $\mathbf{x}_0^{br} \in \mathcal{X}$
6:   **for** round $k = 1, 2, \ldots, K$ **do**           */ BR game
7:    $\mathcal{X}_k^{br} \leftarrow \left\{(x^{i,br}, x^{-i,br}_{k-1}), \text{ such that } x^{i,br} = \arg\max_{x \in \mathcal{X}^{(i)}} \text{UCB}(x, x_{k-1}^{-i,br})\right\}_{i=1}^n$
8:    Play $\mathbf{x}_k^{br} \leftarrow \arg\max_{\mathbf{x}_k \in \mathcal{X}_k^{br}}[\text{UCB}(x_k)]$
9:   **end for**
10:   Collect equilibrium protein sequence $x_{t,m} \leftarrow \mathbf{x}_K^{br}$
11:  **end for**
12:  Select batch of top $B$ equilibrium protein sequences $\{x_{t,i}\}_{i=1}^B$ according to $\text{UCB}(\mathcal{GP}^{t-1}, \beta, \cdot)$.  */ Filtering
13:  Obtain fitness evaluations $y_{t,i} = f(x_{t,i}) + \epsilon_{t,i}, \quad \forall i = 1, \ldots, B$
14:  Update $\mathcal{D}_t \leftarrow \mathcal{D}_{t-1} \cup \{(x_{t,i}, y_{t,i})\}_{i=1}^B$
15:  Posterior update of model $\mathcal{GP}^t$ with $\mathcal{D}_t$
16: **end for**

---

**Algorithm 5** GAMEOPT-HEDGE for Protein Design

---

1: **Input:** GP prior $\mathcal{GP}^0(\mu_0, k(\cdot, \cdot))$, initial data $\mathcal{D}_0 = \{(x_i, y_i = f(x_i) + \epsilon)\}$, protein sites $\mathcal{N}$, batch size $B \in \mathbb{N}$, $M \in \mathbb{N} > B$, parameters $\beta, \eta$.
2: **for** iteration $t = 1, 2, \ldots, T$ **do**
3:  Construct game with reward function $\text{UCB}(\mathcal{GP}^{t-1}, \beta, \cdot) : \prod_{i=1}^n \mathcal{X}^{(i)} \to \mathbb{R}$
4:  **for** $m = 1, 2, \ldots, M$ **do**
5:   Initialize weights $\boldsymbol{w}_1 \leftarrow \frac{1}{W}[1, \ldots, 1] \in \mathbb{R}^{|\mathcal{N}| \times W}$
6:   **for** round $k = 1, 2, \ldots, K$ **do**        */ Simultaneous Hedge
7:    Sample $x_k^i \sim \boldsymbol{w}_1^i, \forall i \in \mathcal{N}$
8:    Set joint strategy $\mathbf{x}_k \leftarrow \{x_k^i\}_{i \in \mathcal{N}}$
9:    **for** player $i \in \mathcal{N}$ **do**           */ Players' payoff
10:     $\ell_{x_k^{-i}} \leftarrow [v(x_k^{j,-i})]_{\forall j \in \mathcal{X}^{(i)}}$, where
      $x_k^{j,-i} = j \cup \{x_k^{i'}\}_{i' \in \mathcal{N} \setminus \{i\}}, \forall j \in \mathcal{X}^{(i)}$
11:     Set $\boldsymbol{w}_{k+1}^i \propto \boldsymbol{w}_k^i \exp(\eta \ell_{x_k^{-i}})$
12:    **end for**
13:   **end for**
14:   Collect equilibrium protein sequence $x_{t,m} \leftarrow \mathbf{x}_K$
15:  **end for**
16:  Select batch of top $B$ equilibrium protein sequences $\{x_{t,i}\}_{i=1}^B$ according to $\text{UCB}(\mathcal{GP}^{t-1}, \beta, \cdot)$.  */ Filtering
17:  Obtain fitness evaluations $y_{t,i} = f(x_{t,i}) + \epsilon_{t,i}, \quad \forall i = 1, \ldots, B$
18:  Update $\mathcal{D}_t \leftarrow \mathcal{D}_{t-1} \cup \{(x_{t,i}, y_{t,i})\}_{i=1}^B$
19:  Posterior update of model $\mathcal{GP}^t$ with $\mathcal{D}_t$
20: **end for**

---

# D EXPERIMENT DETAILS

We set the (hyper)parameters for the experiments as in Table 1.

***GB1(4)*** The dataset (Wu et al., 2016) is fully combinatorial, *i.e.*, encompassing fitness measurements of $20^4$ variants with 4 sites. In this context, each protein site is treated as a player in the cooperative game of GAMEOPT, with $\mathcal{N} = \{1, \ldots, 4\}$. Additionally, we also analyzed the effect of player grouping inspired by *epistasis* phenomenon in protein design and provided the analysis in Appendix E.

---

**Algorithm 6** ITERATIVE BEST RESPONSE-FITNESS (IBR-FITNESS)

---

1: **Input:** Domain $\mathcal{X}$, fitness function $f : \mathcal{X} \to \mathbb{R}$, players $\mathcal{N}$, initial data $\mathcal{D}_0 = \{(x_i, y_i = f(x_i) + \epsilon)\}$, batch size $B \in \mathbb{N}$.
2: $\mathbf{x}_0^{br} \leftarrow$ random joint strategy, $\mathbf{x}_0^{br} \in \mathcal{X}$
3: **for** iteration $t = 1, 2, \ldots, T$ **do**
4:     Randomly selected $B$ players $\in \mathcal{N}$ generates BRs $\{x_{t,i}\}_{i=1}^B$ w.r.t. $\mathbf{x}_{t-1}^{br}$ based on $f(\cdot)$
5:     Obtain evaluations $y_{t,i} = f(x_{t,i}) + \epsilon_{t,i}, \quad \forall i = 1, \ldots, B$
6:     Update $\mathcal{D}_t \leftarrow \mathcal{D}_{t-1} \cup \{(x_{t,i}, y_{t,i})\}_{i=1}^B$
7:     Play $\mathbf{x}_t^{br} \leftarrow \arg\max_{x_{t,i} \in \{x_{t,i}\}_{i=1}^B} y_{t,i}$
8: **end for**
9: **return** $\mathbf{x}_T^\star \leftarrow \arg\max_{(x,y) \in \mathcal{D}_T} y$                                              */ Best-so-far

---

**Algorithm 7** RANDOM

---

1: **Input:** Domain $\mathcal{X}$, $f : \mathcal{X} \to \mathbb{R}$, initial data $\mathcal{D}_0 = \{(x_i, y_i = f(x_i) + \epsilon)\}$, batch size $B \in \mathbb{N}$.
2: **for** iteration $t = 1, 2, \ldots, T$ **do**
3:     Randomly generate batch of $B$ points $\{x_{t,i}\}_{i=1}^B, \forall x_{t,i} \in \mathcal{X}$
4:     Obtain evaluations $y_{t,i} = f(x_{t,i}) + \epsilon_{t,i}, \quad \forall i = 1, \ldots, B$
5:     Update $\mathcal{D}_t \leftarrow \mathcal{D}_{t-1} \cup \{(x_{t,i}, y_{t,i})\}_{i=1}^B$
6: **end for**
7: **return** $\mathbf{x}_T^\star \leftarrow \arg\max_{(x,y) \in \mathcal{D}_T} y$                                                    */ Best-so-far

---

We train the GP surrogate by utilizing a small portion of the dataset, specifically $0.0625\%$, consisting of 100 protein variants. Since existing literature does not provide common ground feature embeddings as representations for the *GB1(4)* variants, we use chemical descriptors (Wu et al., 2019) to extract 60 feature embeddings using a training set of size 1000 protein variants with LASSO method. We apply $k$-fold cross-validation with $k = 18$ different train/test dataset partitions. Following this, we evaluate the performance of our approach over 18 replications. In each replication, we initialize the GP surrogate-based baseline methods with the same initial GP model as our approach. We also use the same initial protein sequence for comparison within that replicate but employ different initial points across replications. We set the starting joint strategy as the protein sequence having the highest log fitness value in the training set. The prior mean of the GP is fixed at 1.0162. For the kernel hyperparameters, 60 lengthscales are defined for each feature dimension and optimized offline at the beginning of a replication; outputscale is set to 0.02169.

***GB1(55)***   We experiment on the non-exhaustive dataset *GB1(55)* that only includes 2-point mutations throughout the entire 55 residues of the *GB1* protein resulting in $535{,}917$ variants (Olson et al., 2014) and consider two settings: 55 and 10 number of players.

***GB1(55)* with 55 Players**   In this context, we treat each protein site as a player in the GAMEOPT, thus, $\mathcal{N} = \{1, \ldots, 55\}$.

As the dataset is not completely combinatorial, we do not have access to measured fitness values for all $20^{55}$ variants. To overcome this, we employ a Deep Neural Network-based (DNN) *oracle* to predict fitness scores using feature embeddings associated with the protein sequences. We again opt to feature embeddings as the representation for categorical input of GP surrogate. Unlike *GB1(4)*, we utilize the `ESM-1v` protein language model from `esm` introduced by Meier et al. (2021), specifically designed for predicting protein variant effects and can be used to extract embeddings. With `ESM-1v`, we represent a sequence through a 1280 dimensional feature embedding vector. We train the *oracle* with supervised learning, using the training set having $(477\,854 \times 1280, 477\,854)$ feature & label pairs. Obtaining the exhaustive version of the *GB1(55)* dataset, we train the GP surrogate using `ESM-1v` feature embeddings of 1000 randomly generated protein variants and corresponding *oracle*-predicted fitness scores for 10 replications.

Table 1: Hyperparameter values.

| Symbol | Hyperparameter | Value |
|---|---|---|
| $T$ | The number of active learning (BO) iterations | 50 |
| $K$ | The number of game rounds | 40 for *GFP*, 200 for *Halogenase* and *GB1(55)*, and 400 for *GB1(4)*; |
| | | 50 for GAMEOPT-IBR in *GB1(55)* & $n = 10$ domain |
| | | 100 for GAMEOPT-IBR in *GB1(55)* & $n = 55$ domain |
| $n = \mid \mathcal{N} \mid$ | The number of players | 3 for *Halogenase*, 4 for *GB1(4)*, 6 and 8 for *GFP*, 10 and 55 for *GB1(55)* |
| $\mid \mathcal{D}_0 \mid$ | The number of samples in training set | 100 for *Halogenase* and *GB1(4)*, and 1000 for *GFP* and *GB1(55)* |
| $\eta$ | Learning rate | Optimized over the set $\{0.1, 0.5, 0.9, 1.0, 1.5, 2.0\}$ |
| $\epsilon$ | Observation noise for each training example | 0.0004 |
| $l$ | RBF kernel lengthscale | Optimized offline |
| $\beta$ | The UCB tuning parameter | 2 |
| $B$ | Batch size per BO iteration | 5 |

***GB1(55)* with 10 players**  To further analyze the performance of GAMEOPT compared to the other baselines, we consider the setting where among the 55 sites, only 10 most significant protein sites can be mutated.

We employ the same protein language model for embeddings and *oracle* to predict fitness scores. However, the choice of 10 players among $\binom{55}{10}$ possibilities is a strategic decision that affects the design performance. For this, we define the significance of a protein site considering the average variation in the fitness scores in the dataset. Concretely, we use Algorithm 8 and select $\mathcal{N} = \{21, 24, 35, 39, 41, 45, 46, 47, 48, 50\}$ sites as the players. We treat the rest of the protein sequence, *i.e.*, sites that do not correspond to players as fixed.

---

**Algorithm 8** COMPUTEMOSTSIGNIFICANTSITES

1: **Input:** Dataset $\mathcal{D} = (x_i, y_i)_{i=1}^N$, players $K$, protein sequence length $L$, amino acids set $\mathcal{A}$.
2: Initialize $players \leftarrow \emptyset$, $site\_score_a^k \leftarrow \emptyset$ and $site\_var^k \leftarrow 0, \forall k \in \{1, \ldots, L\}, a \in \mathcal{A}$
3: **for** each pair $(x_i, y_i) \in \mathcal{D}$ **do**
4:     **for** each site $k \in \{1, \ldots, L\}$ **do**
5:         Set amino acid in site $k$ as $a \leftarrow x_i^k$
6:         Append $site\_score_a^k \leftarrow site\_score_a^k \cup \{y_i\}$
7:     **end for**
8: **end for**
9: $site\_score^k \leftarrow \{site\_score_a^k\}_{a \in \mathcal{A}}, \forall k \in \{1, \ldots, L\}$
10: $site\_var^k \leftarrow stdev(site\_score^k), \forall k \in \{1, \ldots, L\}$
11: **return** $K$ sites having highest $site\_score$ as *players*

---

***Halogenase***  *Halogenase* is a non-exhaustive dataset involving fitness measurements for 605 unique variants. To obtain the complete protein fitness landscape, we again construct an oracle, *i.e.*, an MLP having $R^2 = 0.96$ on a test set and experiment on a setting where each protein site is a player.

***GFP* with 6 players**  The *Aequorea victoria* green-fluorescent protein dataset only includes fitness measurements of $35, 584$ variants corresponding to mixed mutations of some positions on a 238 length sequence. For the fully combinatorial protein fitness landscape, we construct an oracle, *i.e.*, an MLP with test $R^2 = 0.90$ and choose 6 positions: $\mathcal{N} = \{10, 18, 22, 37, 67, 78\}$ that have the largest number of mutations in the original dataset as the players of the GAMEOPT.

***GFP* with 8 players**  To set the players in this setting, we identified 8 positions that have the largest number of mutations in the original dataset: $\mathcal{N} = \{10, 18, 22, 37, 67, 78, 196, 112\}$.

# E ADDITIONAL EXPERIMENTAL RESULTS

## E.1 SAMPLE BATCH DIVERSITY

We also evaluate our method against baselines in terms of performance metric: sampled batch diversity concerning the past. It is measured via the mean Hamming distance of executed points at each BO iteration to the (1) closest initial training point and (2) the proposed point at the previous iteration (pairwise distance).

The results depicted in Figure 4 underscore that GAMEOPT explores at a faster rate compared to baselines except for RANDOM. Notably, even in the initial iterations, GAMEOPT demonstrates the capability to discover points beyond the trust region of its GP prior. Furthermore, it consistently upholds the sampled batch diversity compared to previously executed strategies. As also illustrated in Figure 3, GAMEOPT explores effectively at the beginning and gradually converges to a region conducive to exploitation. This enhanced exploration across the search space contributes to its outperforming performance in identifying high fitness-valued protein sequences.

On the other hand, the exploration strategy employed by RANDOM relies on the generation of $B$ best responses through random selection, a method that does not consistently ensure a diverse sampled batch in the input space. Furthermore, IBR-FITNESS shows a moderate sampled batch diversity concerning the past, attributed to its more exploitative nature—specifically, the sampling of $B$ best responses based on true log fitness values in comparison to other baseline methods. While PR manages to maintain a diverse sampled batch concerning the past in the context of *GB1(4)*, its performance falters when applied to other settings. Additionally, the sampling process of PR involves computing the expected UCB across all potential strategy combinations of players, making its performance, and consequently its sampled batch diversity, highly reliant on an accurate estimate of this expectation.

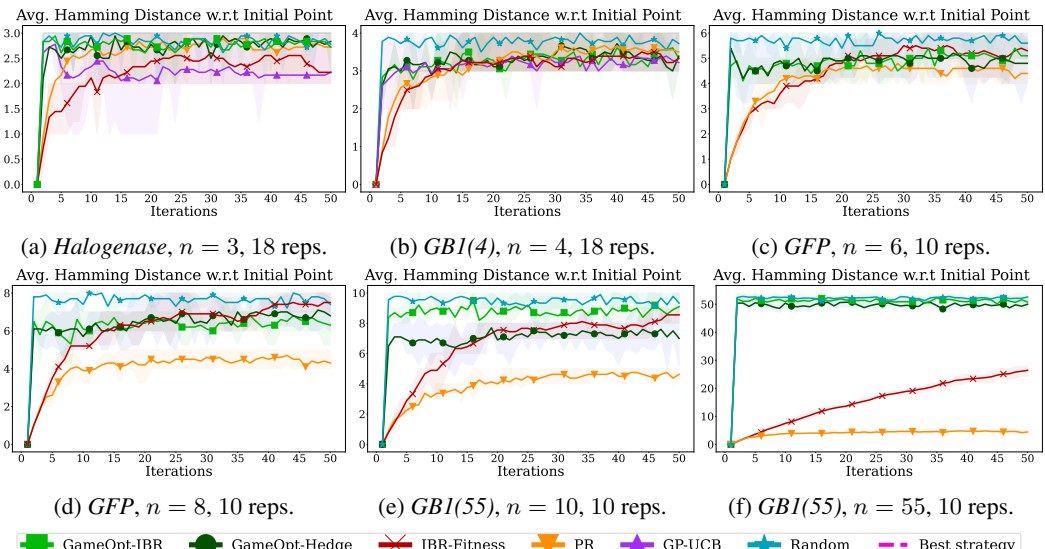

(a) *Halogenase*, $n = 3$, 18 reps.  (b) *GB1(4)*, $n = 4$, 18 reps.  (c) *GFP*, $n = 6$, 10 reps.

(d) *GFP*, $n = 8$, 10 reps.  (e) *GB1(55)*, $n = 10$, 10 reps.  (f) *GB1(55)*, $n = 55$, 10 reps.

GameOpt-IBR  GameOpt-Hedge  IBR-Fitness  PR  GP-UCB  Random  Best strategy

Figure 4: Performance results for the sampled batch diversity w.r.t past measured via mean Hamming distance between the executed variant and the closest initial point from the training set, under batch size $B = 5$. Each point on each line is the average of multiple replications initiated with different training sets having 100 variants for *GB1(4)* & *Halogenase* and 1000 for *GB1(55)* & *GFP*. Similarly, error bars are interquartile ranges averaged over replications. In all experiments, GAMEOPT explores significantly faster than the baseline methods.

### E.2 COMPARISON WITH GP-UCB

We further analyzed the saturating behavior exhibited by GP-UCB in the *GB1(4)* setting. As depicted in Figure 5, our investigation focused on the influence of the initial GP surrogate model, considering different training set sizes, specifically with 100 and 500 training points.

Our findings underscore that the efficacy of GP-UCB heavily relies on the quality of the initial GP surrogate model. Particularly, an initial GP surrogate trained with only 100 data points proves insufficient for GP-UCB to effectively identify high-log fitness protein sequences. Given that GP-UCB optimizes the UCB globally and selects the $B$ best points in each iteration, the limited informativeness of sampled batch points under this GP surrogate constrains the algorithm. Therefore, GP-UCB ends up converging to a point where further improvement is impeded. In contrast, employing a potentially more informative GP model with 500 training points enables GP-UCB to perform comparably to GAMEOPT. Our proposed approach, however, exhibits robustness by overcoming the constraints associated with a model initialized with limited data. Through computing evaluation points as the equilibria of cooperative game-playing, it consistently gathers diverse and informative batches to guide the GP, thereby enhancing sample efficiency.

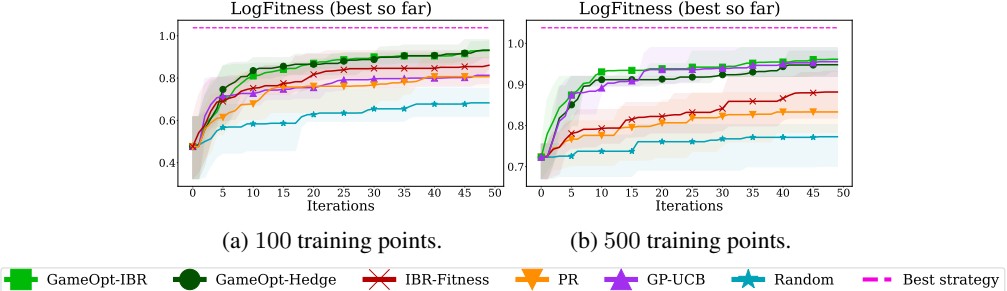

(a) 100 training points.    (b) 500 training points.

Figure 5: The effect of training set size on the performance under setting *GB1(4)*, $n = 4$, 18 reps. GP-UCB mitigates saturating behavior when leveraging a more informative initial GP surrogate model. In contrast, GAMEOPT showcases resilience in overcoming the limitations associated with a GP model trained with a limited amount of data.

### E.3 COMPARISON WITH DISCRETE LOCAL SEARCH METHODS

We further compared GAMEOPT against Discrete Local Search baselines DLS and DLS_BEST (Balandat et al., 2020) that perform a discrete local search by exploring k-Hamming distance neighborhood. While DLS employs random initialization, DLS_BEST starts the local search from the best-discovered sequence. In our experiments, we set their neighborhood size as 2-Hamming distance and let the baselines greedily select $B$ sequences at each iteration. For a fair comparison against DLS_BEST, we also run the best-discovered sequence initialization version of our approach, called GAMEOPT-IBR_BEST.

We remark that DLS and DLS_BEST can be seen as *constrained* versions of GAMEOPT that require a prior definition of the neighborhood (thus, unlike our approach, they require a notion of distance too). Moreover, being *centralized*, they are subject to a higher computational complexity which grows *exponentially* with the neighborhood size. For a sequence of length $n$ (players) with $| \mathcal{X}^{(i)} |= d$ many amino acid choices, GAMEOPT reduces the intractable optimization of acquisition function (i.e., $O(d^n)$) to $O(nd)$ complexity. Instead, DLS & DLS_BEST search over $k$-distance neighborhoods yielding $O(C(n, k)d^k)$ complexity, where $C$ denotes the combination operation and $k$ is the Hamming distance. We observe GAMEOPT performs comparably to DLS baselines (see Figures 6, 7, 8, and additional analyses below), but requiring a significantly lower compute cost, as shown in Appendix E.4.

### E.4 Computational costs

We report the total compute amount (measured in hours) for the experiments with $T = 50$ BO iterations in Table 2. We conduct the experiments on an internal compute cluster equipped with NVIDIA A100 80 GB tensor-core GPUs. For smaller domains (with short protein sequences), we were able to execute replications concurrently without encountering memory errors. However, in configurations necessitating the generation of embeddings for numerous evaluation points with longer protein sequences in each iteration, parallel execution was not feasible, thus necessitating a per-replication computation time report. Specifically, we present the total compute amount for *all* replications across domains *Halogenase*, *GB1(4)*, and *GFP*; whereas we report *per replication* compute amount accompanied by its standard deviation for the two largest *GB1(55)* domains.

Referring to Table 2, GameOpt variations provide *tractable* acquisition function optimization. Their computational demand predominantly hinges on the number of sequences they evaluate per BO iteration. This is also influenced by the number of game rounds–which we intentionally set to a higher value to guarantee convergence to game equilibrium (see Table 1 for the exact values). Nevertheless, our observations reveal that game equilibrium is often reached well before the rounds are completed. Consequently, the computational times reported for GameOpt can be considered to be within the *worst-case scenario*.

Table 2: Total compute amount (in hours) required for experiments with $T = 50$ BO iterations. Each entry with $\pm$ represents the average *per-replication* computation time, accompanied by their standard deviation, whereas the others show the total compute time *for all* replications. We run 18 replications for *Halogenase*, *GB1(4)*, and 10 replications for the other domains. In all domains, particularly for the larger search spaces, GameOpt provides tractable acquisition function optimization.

| Method | Domain | | | | | |
|---|---|---|---|---|---|---|
| | Halogenase | GB1(4) | GFP, $n = 6$ | GFP, $n = 8$ | GB1(55), $n = 10$ | GB1(55), $n = 55$ |
| GameOpt-Ibr | **0.19** | **0.30** | **1.02** | **2.74** | **1.61 $\pm$ 0.46** | **11.25 $\pm$ 3.19** |
| GameOpt-Hedge | 2.44 | 1.95 | 2.05 $\pm$ 0.02 | 2.60 $\pm$ 0.09 | 3.85 $\pm$ 0.39 | 20.13 $\pm$ 1.04 |
| GP-UCB | 0.63 | 29.56 | - | - | - | - |
| DLS | 0.29 | 1.10 | 2.90 $\pm$ 0.33 | 4.94 $\pm$ 0.24 | 4.10 $\pm$ 2.36 | 102.86 $\pm$ 27.42 |

### E.5 Additional analyses

We performed further analyses to assess the effectiveness of our GameOpt framework against baselines in terms of: (1) fraction of global optima discovered, (2) fraction of solutions found above a fitness threshold, (3) cumulative maximum for sequences proposed and (4) mean pairwise Hamming distance between proposed sequences.

We evaluated the fraction of global optima discovered under the *GB1(4)* setting, as depicted in Table 3. For this fully combinatorial dataset, we identify the global optimizer sequence by exhaustive search. Our findings revealed that GameOpt-IBR successfully identified the global optimum sequence in 33.33% of cases out of 18 replications, highlighting its superior convergence against baselines.

Additionally, given the difficulty of identifying global optima in non-fully combinatorial datasets, we examined the fraction of optima found above a fitness threshold, $f_\tau = 0.8$. Table 4 demonstrates that across all problem domains (excluding best cumbersome initialization versions), the GameOpt framework consistently samples batches containing a higher number of optimal sequences compared to baseline methods. Its effectiveness is highlighted more as the search space size gets higher. In the most complex domain, *GB1(55)* with 55 players, its best cumbersome initialization version performs comparably to DLS_Best. However, it is essential to also acknowledge the comparison w.r.t the computational complexity associated with that configuration as discussed in Appendix E.4.

Regarding the cumulative maximum for proposed sequences (see Table 5), our framework demonstrates notable performance by consistently proposing protein variants with higher fitness values. Moreover, its superior convergence speed, illustrated in Figure 6, underscores its effectiveness against baselines including DLS. The discrete local search with best initialization, DLS_Best, converges relatively slower, particularly in small domains, yet performs comparably against GameOpt_Best in finding high log fitness valued variants. However, it does not provide tractable optimization as detailed in Appendix E.4.

Furthermore, we analyzed the performance of methods considering the mean pairwise Hamming distance between the sequences proposed at each BO iteration (see Table 6 and Figure 7) which is an indicator for sampled batch diversity. The results indicate that GAMEOPT explores moderately, however, it balances exploration and exploitation to prevent over-exploring seen in RANDOM and DLS, as well as over-exploiting observed in GAMEOPT_BEST, DLS_BEST and PR.

Lastly, the mean Hamming distance between the proposed variant and the closest initial point from the training set (see Figure 8) showcases that GAMEOPT explores the solution space faster than the baseline methods, except for RANDOM and DLS.

Table 3: Fraction of global optima found for the *GB1(4)* dataset. Each entry is the average of 18 replications. GAMEOPT variations are able to sample global optimum sequence (AHCA) more frequently compared to other baselines. Entries of the outperforming methods are denoted in bold. The results show that GAMEOPT-IBR converges to the best strategy more frequently compared to the baselines.

| Method | % best strategy (AHCA) found |
|---|---|
| GAMEOPT-IBR | **33.33** |
| GAMEOPT-HEDGE | 16.67 |
| IBR-FITNESS | 16.67 |
| PR | 0.00 |
| GP-UCB | 0.00 |
| RANDOM | 0.00 |
| DLS | 0.00 |
| DLS_BEST | 11.11 |
| GAMEOPT-IBR_BEST | 11.11 |

Table 4: Percentage of computed candidates that are above a threshold of $f_\tau = 0.8*(\text{maximum fitness value})$. Each entry is the average of 18 replications for *Halogenase* and *GB1(4)* settings, and 10 replications for the others. Entries of the outperforming methods are denoted in bold. Results indicate the effectiveness of GAMEOPT variations on sampling more optima than the baseline methods.

| Method | Domain | | | | | |
|---|---|---|---|---|---|---|
| | Halogenase | GB1(4) | GFP, $n = 6$ | GFP, $n = 8$ | GB1(55), $n = 10$ | GB1(55), $n = 55$ |
| GAMEOPT-IBR | **100.00** | **21.20** | **28.52** | 50.96 | **11.60** | 1.96 |
| GAMEOPT-HEDGE | 94.44 | 14.62 | 26.68 | **61.72** | 9.48 | 2.12 |
| IBR-FITNESS | 72.22 | 7.46 | 18.96 | 35.32 | 0.36 | 0.00 |
| PR | 38.89 | 2.05 | 15.96 | 15.32 | 0.00 | 0.04 |
| GP-UCB | 72.22 | 4.82 | - | - | - | - |
| RANDOM | 0.00 | 0.29 | 0.76 | 1.60 | 0.00 | 0.00 |
| DLS | 94.44 | 7.46 | 22.60 | 36.48 | 0.16 | 0.00 |
| DLS_BEST | 88.89 | 19.88 | 16.80 | 45.92 | 2.64 | **52.24** |
| GAMEOPT-IBR_BEST | 44.44 | 17.40 | 12.32 | 42.64 | 1.68 | 48.64 |

Table 5: Cumulative maximum for sequences proposed at the end of the BO iterations. Each entry is the average of 18 replications for *Halogenase* and *GB1(4)* settings, and 10 replications for the others. Entries of the outperforming methods are denoted in bold. GAMEOPT variations show superior performance in proposing higher fitness-valued protein variants.

| Method | Domain | | | | | |
|---|---|---|---|---|---|---|
| | Halogenase | GB1(4) | GFP, $n = 6$ | GFP, $n = 8$ | GB1(55), $n = 10$ | GB1(55), $n = 55$ |
| GAMEOPT-IBR | **3.51** | 0.93 | **6.17** | **6.37** | 1.94 | 4.72 |
| GAMEOPT-HEDGE | 3.45 | **0.94** | 6.15 | 6.34 | 1.69 | 4.66 |
| IBR-FITNESS | 3.10 | 0.86 | 6.08 | 6.21 | 0.97 | 2.99 |
| PR | 2.08 | 0.81 | 6.06 | 6.07 | 0.90 | 1.18 |
| GP-UCB | 2.98 | 0.82 | - | - | - | - |
| RANDOM | 1.15 | 0.68 | 5.92 | 5.97 | -0.08 | -0.66 |
| DLS | **3.51** | 0.89 | 6.09 | 6.14 | 1.70 | 1.35 |
| DLS_BEST | 3.40 | 0.93 | 6.15 | 6.33 | **2.11** | **7.87** |
| GAMEOPT-IBR_BEST | 2.28 | 0.93 | 6.14 | 6.33 | 1.91 | 7.25 |

Table 6: Mean pairwise Hamming distance between the sequences proposed at each iteration. Each entry is the average of 18 replications for *Halogenase* and *GB1(4)* settings, and 10 replications for the others. In all problem domains, RANDOM baseline consistently samples a rather diverse batch of evaluation points w.r.t. past proposed variants. Although this shows enhanced exploration, one drawback is the lack of exploitation. Hence, as illustrated in Figure 3, GAMEOPT balances these two successfully and shows a moderate sampled batch diversity.

| Method | Domain | | | | | |
|---|---|---|---|---|---|---|
| | Halogenase | GB1(4) | GFP, $n = 6$ | GFP, $n = 8$ | GB1(55), $n = 10$ | GB1(55), $n = 55$ |
| GAMEOPT-IBR | 1.40 | 2.94 | 3.08 | 4.11 | 7.32 | 50.93 |
| GAMEOPT-HEDGE | 2.13 | 2.91 | 2.93 | 4.23 | 3.98 | 45.89 |
| IBR-FITNESS | 0.86 | 0.88 | 0.85 | 0.89 | 0.86 | 0.89 |
| PR | 1.61 | 1.57 | 1.61 | 1.67 | 1.69 | 1.60 |
| GP-UCB | 0.77 | 2.41 | - | - | - | - |
| RANDOM | 2.85 | 3.8 | 5.69 | 7.64 | 9.51 | 52.34 |
| DLS | 1.13 | 3.36 | 5.13 | 7.13 | 9.38 | 52.19 |
| DLS_BEST | 0.68 | 2.56 | 1.18 | 2.36 | 1.55 | 3.57 |
| GAMEOPT-IBR_BEST | 0.33 | 2.64 | 1.48 | 2.78 | 0.57 | 3.12 |

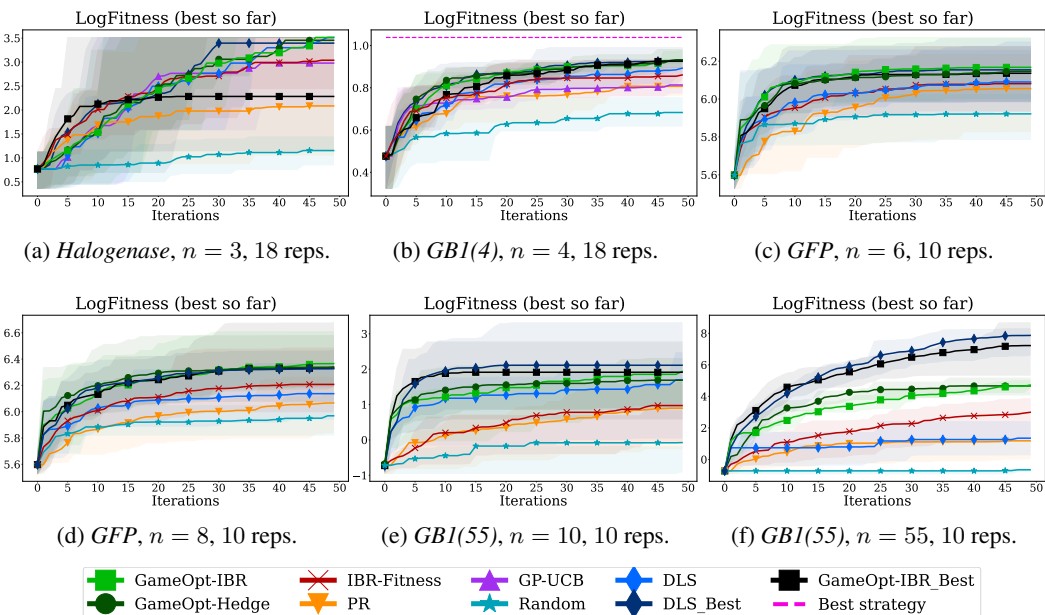

(a) *Halogenase*, $n = 3$, 18 reps.   (b) *GB1(4)*, $n = 4$, 18 reps.   (c) *GFP*, $n = 6$, 10 reps.

(d) *GFP*, $n = 8$, 10 reps.   (e) *GB1(55)*, $n = 10$, 10 reps.   (f) *GB1(55)*, $n = 55$, 10 reps.

Figure 6: Convergence speed of methods in terms of log fitness value of the best-so-far protein throughout BO iterations, under batch size $B = 5$. Each point is the average of multiple replications initiated with different training sets having 100 protein variants for *Halogenase* and *GB1(4)*, and 1000 protein variants for the rest of the problem domains. Similarly, error bars are interquartile ranges averaged over replications.

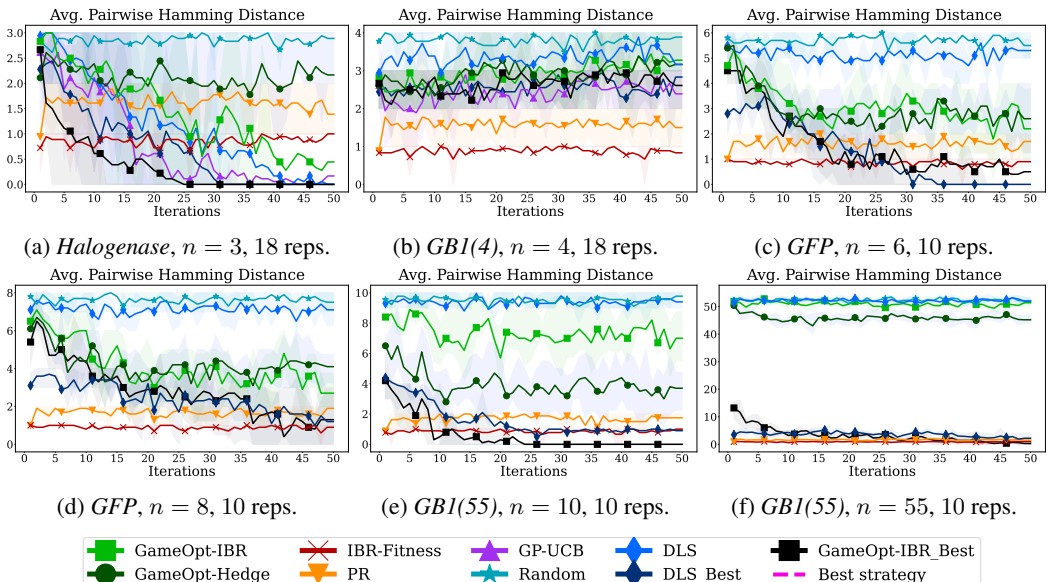

(a) *Halogenase*, $n = 3$, 18 reps.  (b) *GB1(4)*, $n = 4$, 18 reps.  (c) *GFP*, $n = 6$, 10 reps.

(d) *GFP*, $n = 8$, 10 reps.  (e) *GB1(55)*, $n = 10$, 10 reps.  (f) *GB1(55)*, $n = 55$, 10 reps.

Figure 7: Sampled batch diversity w.r.t past measured via mean Hamming distance between the executed variant and the proposed variant from the previous iteration (pairwise distance), under batch size $B = 5$. Each point on each line is the average of multiple replications initiated with different training sets having 100 variants for *GB1(4)* & *Halogenase* and 1000 for *GB1(55)* & *GFP*. Similarly, error bars are interquartile ranges averaged over replications. In all experiments GAMEOPT consistently samples a rather diverse batch of evaluation points w.r.t. past proposed variants. This enhanced exploration of the search space contributes to its strong performance compared to the baseline methods.

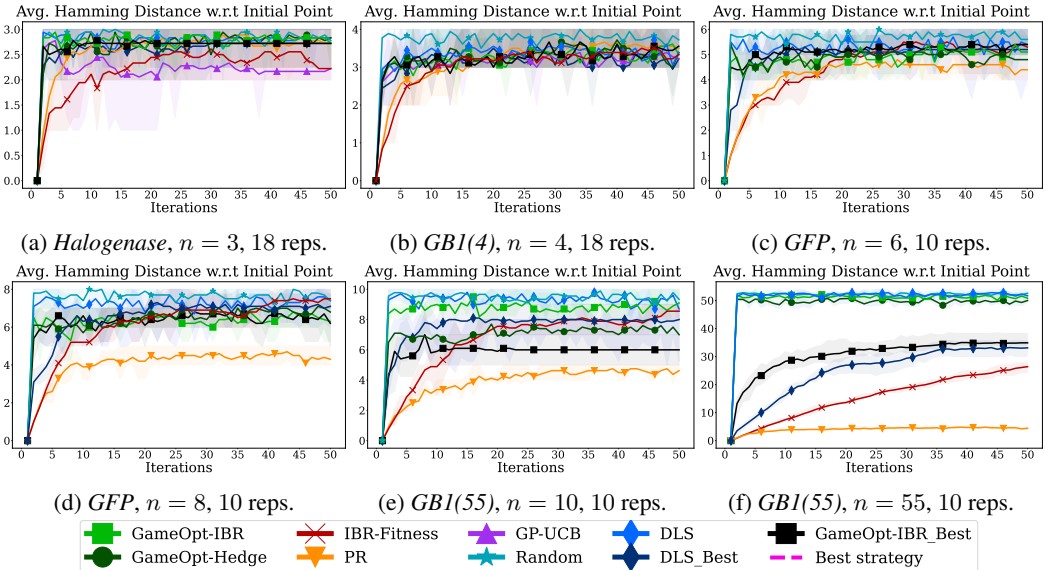

(a) *Halogenase*, $n = 3$, 18 reps.  (b) *GB1(4)*, $n = 4$, 18 reps.  (c) *GFP*, $n = 6$, 10 reps.

(d) *GFP*, $n = 8$, 10 reps.  (e) *GB1(55)*, $n = 10$, 10 reps.  (f) *GB1(55)*, $n = 55$, 10 reps.

Figure 8: Performance results for the sampled batch diversity w.r.t past measured via mean Hamming distance between the executed variant and the closest initial point from the training set, under batch size $B = 5$. Each point on each line is the average of multiple replications initiated with different training sets having 100 variants for *GB1(4)* & *Halogenase* and 1000 for *GB1(55)* & *GFP*. Similarly, error bars are interquartile ranges averaged over replications. In all experiments, GAMEOPT explores faster than the baseline methods, except for RANDOM and DLS.

### E.6 EXPLORING PLAYERS' GROUPING

Until this juncture, we have exclusively examined scenarios where a GAMEOPT player is responsible for only a single site within the protein sequence. In light of the *epistasis* (Phillips, 2008) phenomenon in protein design, which underscores how the effect of a mutation on fitness can be influenced by the presence of other mutations within the same protein, we now explore the concept of grouping protein sites together, *i.e.*, having players being responsible for more than one site. This is because modeling protein sites independently may yield different fitness outcomes than finding equilibria among groups of several sites. To this end, we conduct a preliminary investigation into whether this phenomenon alters GAMEOPT's performance.

We experiment on *GB1(4)* with $\{0, 1, 2, 3\}$ protein sites and $\mathcal{N} = \{1, 2\}$ players, considering 3 possible player & site groupings: $\{(01, 23), (02, 13), (03, 12)\}$. For instance, setting $(01, 23)$ means that the first player is responsible for sites $\{0, 1\}$ and the other one for $\{2, 3\}$.

Our evaluations with GAMEOPT-IBR and GAMEOPT-HEDGE using the same performance measures (Figures 9 and 10) showed that there is no significant performance difference between individual players and grouping settings as they all discover the high log fitness valued protein variants at a similar rate while collecting batches of diverse evaluation points. Nevertheless, an in-depth examination of this phenomenon on larger datasets remains a subject for future investigation.

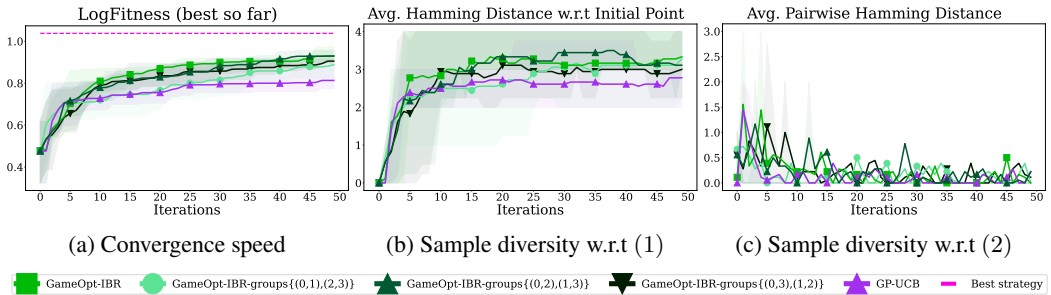

Figure 9: GAMEOPT-IBR performance for player grouping, under *GB1(4)* setting, 18 reps. There is no significant performance difference between individual players and player grouping settings under this domain.

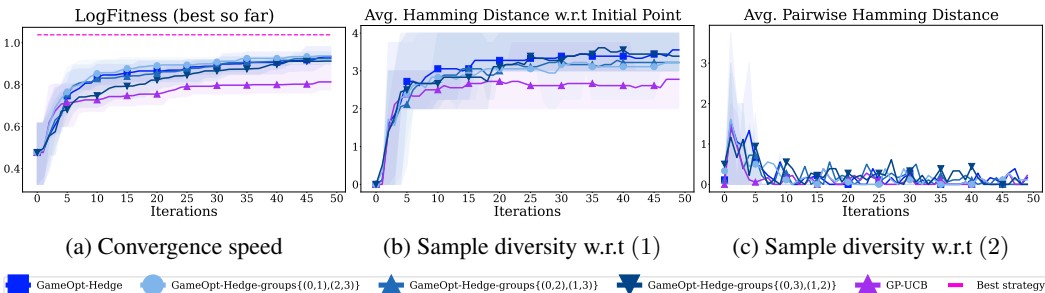

Figure 10: GAMEOPT-HEDGE performance for player grouping, under *GB1(4)* setting, 18 reps. No significant performance difference exists between individual players and player grouping settings under this domain.

### E.7 EXPLORING PLAYERS' WITH DIFFERENT ACTION SET SIZES (DIMENSIONS)

In our main experiments presented in Section 5.4, as well as the grouping of players explored in Appendix E.6, we exclusively examined scenarios where GAMEOPT players have an equal number of actions, i.e. $|\mathcal{X}^{(i)}|$ are same for all $i$. In the context of protein design, this corresponds to the setting where players are responsible for an equal number of sites. However, GAMEOPT can also be applied to the setting where players have *different* numbers of actions, i.e. $|\mathcal{X}^{(i)}|$ differ among players.

To demonstrate the *generalizability* of GAMEOPT to such settings, we further experiment on settings with player groupings, where each group is responsible for a different number of protein sites. Particularly, we consider site groupings: $\{(0,12),(1,02),(2,01)\}$ for the *Halogenase* problem domain. Whereas we consider groupings: $\{(0,123),(1,023),(2,013),(3,012)\}$ for the *GB1(4)* domain. For instance, the setting $(0,12)$ means that the first player is responsible for site $\{0\}$ and the other one for $\{1,2\}$, which makes their action size as $\mathcal{X}^{(i=1)} = 20, \mathcal{X}^{(i=2)} = 20^2$.

The experiment results presented in Figure 11 show that there is no significant performance difference between individual players and player grouping settings with varying action set sizes, however, in most of the settings GAMEOPT groupings outperform GP-UCB baseline. This clearly demonstrates the applicability of GAMEOPT framework under problem domains with unequal action set sizes.

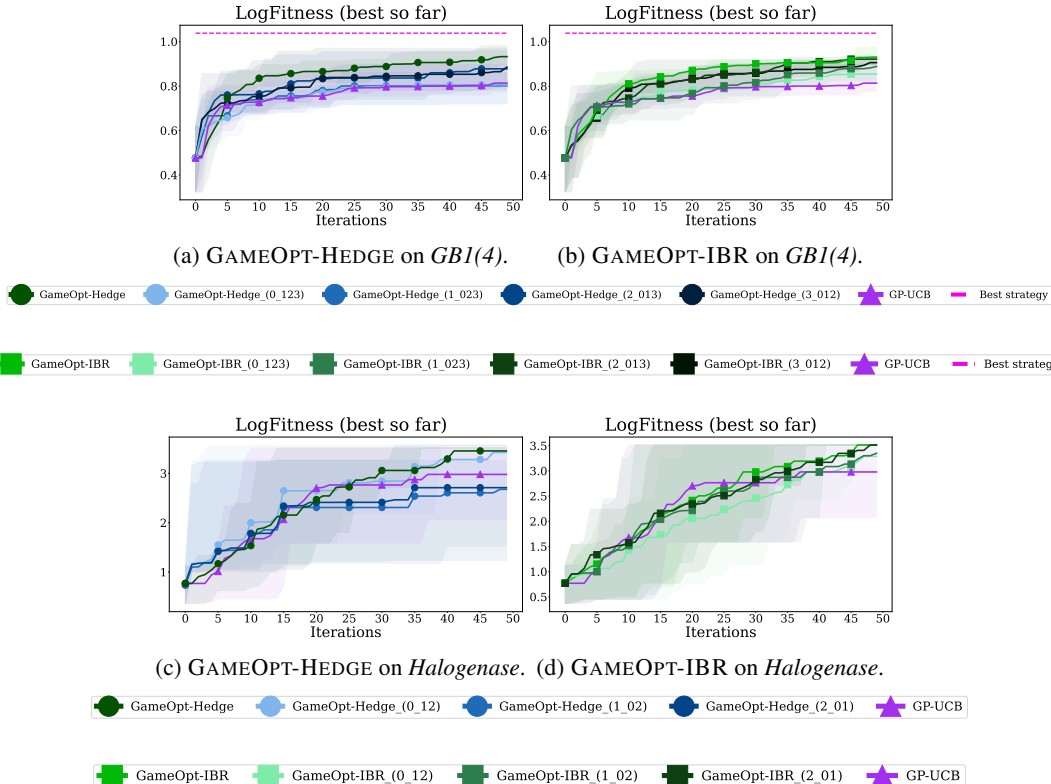

Figure 11: GAMEOPT-HEDGE and GAMEOPT-IBR performance under player groupings with different action sizes, on *GB1(4)* and *Halogenase* domains, 18 reps. There is no significant performance difference between individual players and player grouping settings, however, in most of the settings GAMEOPT groupings outperform the GP-UCB baseline. The results support that GAMEOPT is also effective under problem settings with varying action (dimension) sizes.

### E.8   BATCH SIZE ABLATIONS

We present additional results on the *GB1(4)* dataset using higher batch sizes, $B = 10$ and $B = 50$. The results in Figure 12 show that GAMEOPT variations still outperform baselines by discovering higher log fitness-valued protein sequences at a faster rate due to sampling diverse sets of batches. When batch size increases, GAMEOPT-HEDGE becomes dominant in discovering the best protein sequence. This shows that irrespective of the batch size, GAMEOPT is effective in large combinatorial BO settings.

We further present additional analysis using more restrictive batch sizes, $B = 1$ and $B = 3$ on the *GB1(4)* and *Halogenase* datasets. The results in Figure 13 demonstrate that even under a more restricted setting on both problem domains, GAMEOPT variations achieve superior performance compared to the baselines. Although initially surpassed by GP-UCB under batch size $B = 1$,

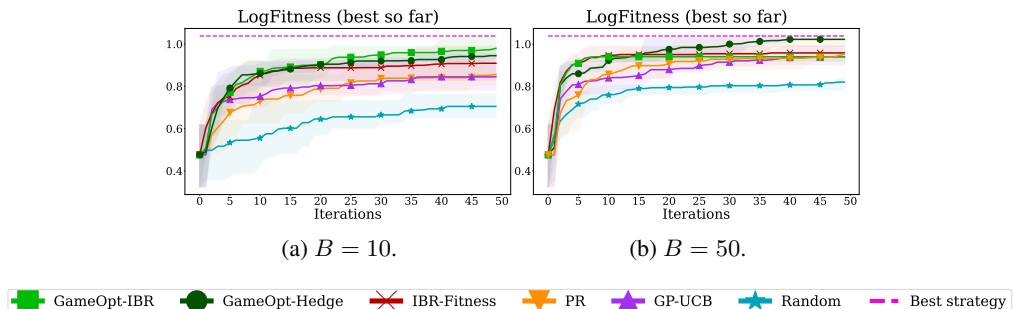

Figure 12: Performance comparison on the *GB1(4)* dataset using batch sizes $B = 10$ and $B = 50$.

GAMEOPT's better exploration compared to the GP-UCB's exploitative behavior avoids ending up at points with lower fitness values. As the batch size increases, GAMEOPT's optimistic game approach benefits from parallelism and collects diverse local optima, hence, GAMEOPT performs significantly better against baselines.

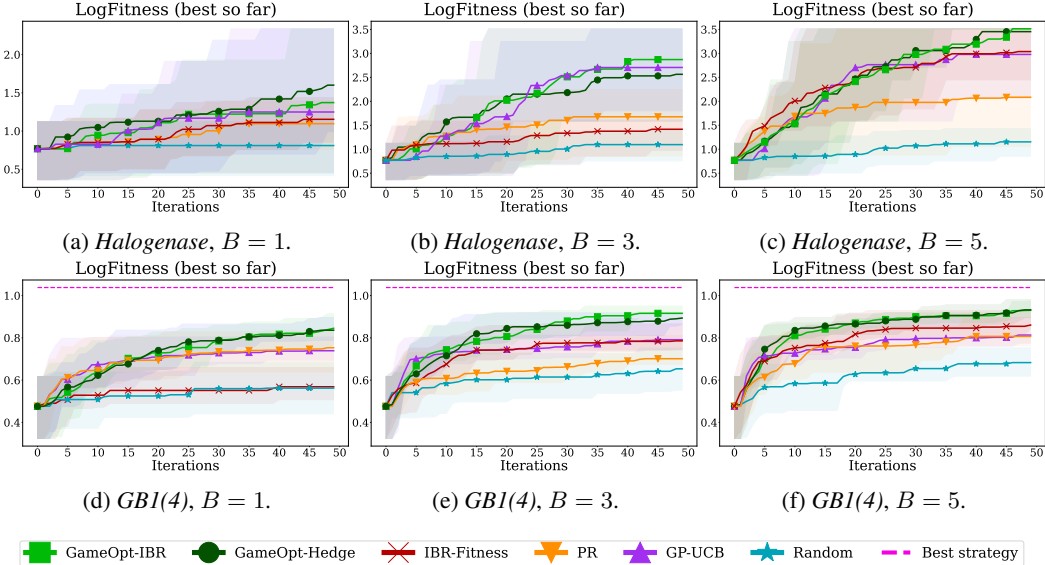

Figure 13: Performance comparison on the *Halogenase* and *GB1(4)* datasets using batch sizes $B = \{1, 3, 5\}$. The results show that GAMEOPT outperforms baselines even under more restrictive batch size settings. Although initially surpassed by GP-UCB under batch size $B = 1$, GAMEOPT's better exploration compared to the GP-UCB's exploitative behavior avoids ending up at points with lower fitness values.

### E.9 LEARNING RATE ($\eta$) ABLATIONS FOR GAMEOPT-HEDGE

To select the learning rate ($\eta$) hyperparameter for the GAMEOPT-HEDGE algorithm, we performed a hyperparameter sweep and tuned it accordingly. Figure 14 illustrates this process, showcasing the impact of different $\eta$ values on the game convergence. Based on this, we selected the optimal performing value. In particular, Figure 14b shows how varying $\eta$ influences the convergence to different equilibria at a BO iteration $t$. The results demonstrate that the chosen $\eta$ facilitates equilibrium convergence within a finite number of rounds, ensuring practical game convergence.

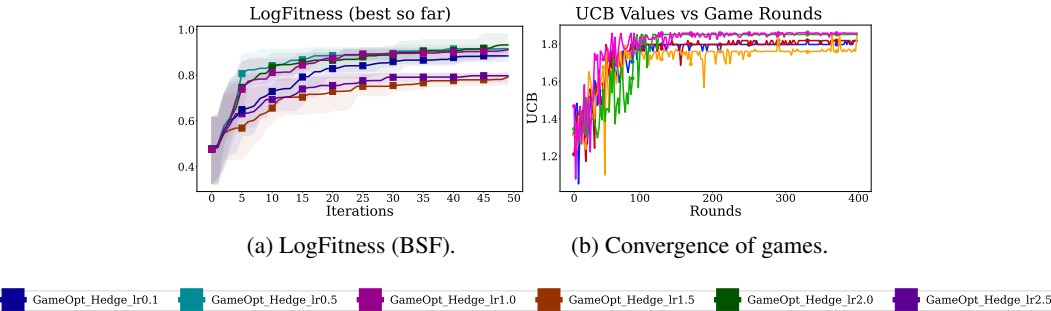

(a) LogFitness (BSF).   (b) Convergence of games.

Figure 14: Performance of GAMEOPT-HEDGE under different learning rate ($\eta$) values. The final value is set to ensure equilibrium convergence within a given number of rounds. As an example, we show in (14b) the convergence of different games to equilibria when $\eta = 2.0$.

### E.10 UCB VALUES OF THE SAMPLED BATCHES

As discussed in Sections 1 and 5.5, the novel insight and the main cause of benefit of the GAMEOPT framework lies in its ability to provide efficient (*tractable*) acquisition function optimization by adopting a game-theoretical perspective.

When comparing the methods based on the UCB values over BO iterations in the *GB1(4)* domain, as demonstrated in Figure 15, we observe that GAMEOPT initially selects points with lower UCB values compared to GP-UCB. However, in later iterations, GAMEOPT identifies points with higher UCB values. This improvement is driven by the framework's leverage of equilibrium points, leading to superior exploration of the search space and ultimately more efficient optimization.

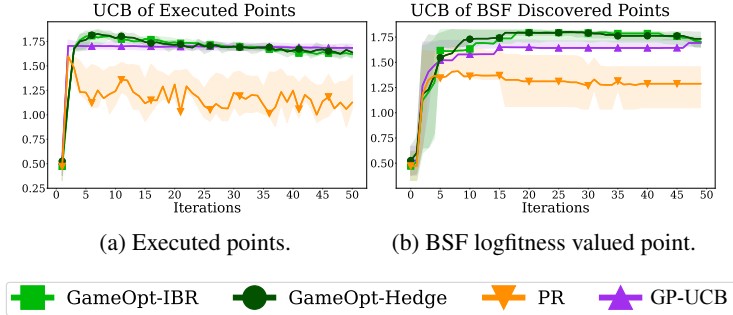

(a) Executed points.   (b) BSF logfitness valued point.

Figure 15: UCB value of the (a): executed point (max UCB), and (b): best-so-far (BSF) logfitness valued point over iterations under *GB1(4)* domain. Although initially GAMEOPT variations executed points with smaller UCB values compared to GP-UCB, its better exploration helps to identify higher UCB valued points quickly.

### E.11 EXPECTED IMPROVEMENT ACQUISITION FUNCTION

Although GAMEOPT is designed using the UCB acquisition function with a sample complexity guarantee, we provide a further empirical analysis across GP-based methods using a different acquisition function: expected improvement (EI) (Jones et al., 1998). As demonstrated by the results on the *GB1(4)* dataset in Figure 16, GAMEOPT variations show superior performance compared to other GP-based baselines. They sample more diverse batches, as given in plots for (16c) sample batch diversity with respect to past and (16d) previously executed points.

### E.12 COMPARISON WITH LATENT SPACE OPTIMIZERS

Our baselines involve acquisition function optimizers which *directly* operate on large *combinatorial* search spaces. While there is a body of work employing latent space optimizers (Gómez-Bombarelli et al., 2018; Tripp et al., 2020; Deshwal & Doppa, 2021; Maus et al., 2022; Stanton et al., 2022), our

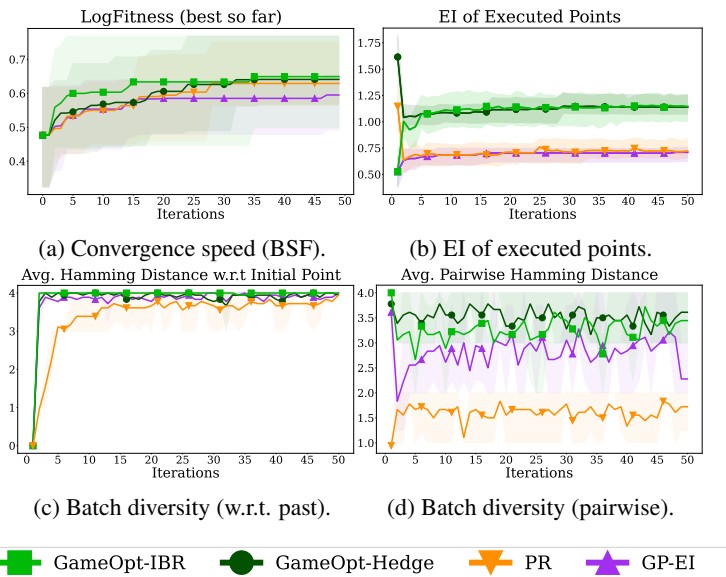

(a) Convergence speed (BSF).  (b) EI of executed points.

(c) Batch diversity (w.r.t. past).  (d) Batch diversity (pairwise).

GameOpt-IBR  GameOpt-Hedge  PR  GP-EI

Figure 16: Performance comparison of GP-based methods on the *GB1(4)* dataset using expected improvement (EI) acquisition function and batch size $B = 5$. In all experiments GAMEOPT with IBR and HEDGE subroutines discover better and more diverse protein sequences at a much faster rate.

primary focus in this study is to *tractably* optimize the acquisition function directly on combinatorial search spaces, addressing the challenge of exhaustive exploration.

However, to provide insight, we compare GAMEOPT against Naïve LSBO-(L-BFGS-B) with second-order gradient-based optimizer (Gómez-Bombarelli et al., 2018) and LADDER (Deshwal & Doppa, 2021) methods. The comparison, presented in Figure 17, is performed using the (17a,17c) RBF kernel and (17b,17d) a structure-coupled kernel. The structure-coupled kernel is designed using an RBF kernel and a sub-sequence string kernel (Moss et al., 2020). Results indicate that GAMEOPT variations perform significantly better than the considered latent space optimizers. A notable limitation of latent space optimizers is their dependence on a decoder. In our experiments, we trained a transformer-based decoder using ESM-1v feature embeddings (Meier et al., 2021), employing beam search for sequence generation. In contrast, our GAMEOPT approach does not require such an additional decoder and *directly* optimizes over the sequence space *efficiently*.

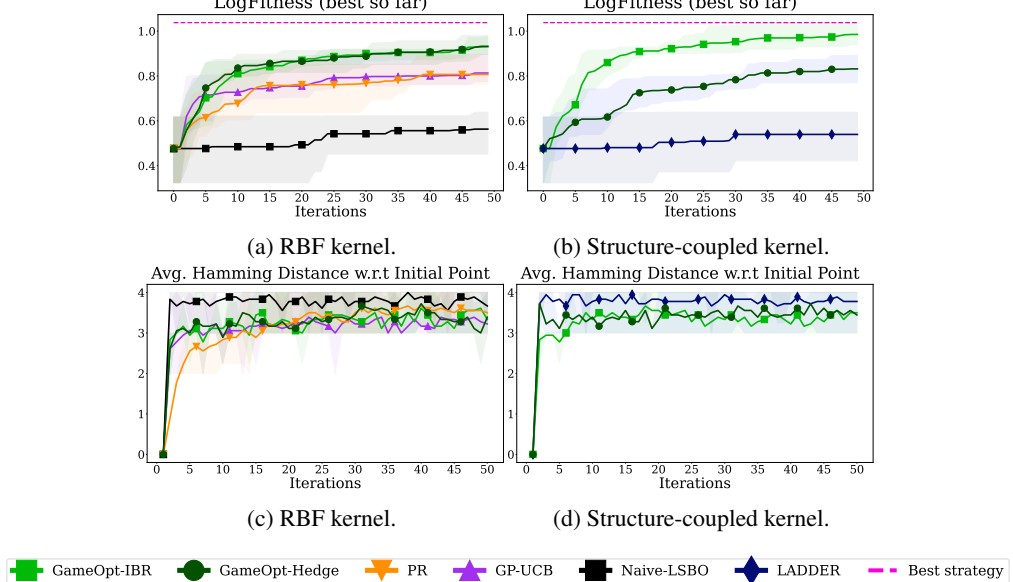

(a) RBF kernel.

(b) Structure-coupled kernel.

(c) RBF kernel.

(d) Structure-coupled kernel.

Figure 17: Performance comparison of GAMEOPT against latent space optimizers: Naïve LSBO-(L-BFGS-B) (Gómez-Bombarelli et al., 2018) and LADDER (Deshwal & Doppa, 2021) methods under *GB1(4)* domain with batch size $B = 5$, using kernels: RBF and structure-coupled kernel (Moss et al., 2020). Latent space optimizers perform poorly mainly because they rely on a decoder, which GAMEOPT variations eliminate and directly (tractably) optimize over combinatorial space.

