# OpenReview forum: "Optimistic Games for Combinatorial Bayesian Optimization with Application to Protein Design"
_ICLR.cc/2025/Conference — ICLR 2025 Poster_

### Official Review · Reviewer_XU81 · 2024-10-23

**Soundness:** 2
**Presentation:** 2
**Contribution:** 2
**Rating:** 3
**Confidence:** 3

**Summary:**

This paper tackles scalability challenges in combinatorial black-box optimization tasks, specifically when each dimension contains the same number of discrete points. While this assumption is not universally applicable in combinatorial Bayesian optimization, it is relevant in specific domains, such as protein engineering. The authors introduce GameOpt, a novel approach that leverages the uniform structure of the discrete search space. By reframing the combinatorial optimization problem as an $n$-player cooperative game, the method aims to identify an $\epsilon$-Nash equilibrium at each iteration. This transformation enables the application of scalable solvers and polynomial-time equilibrium-finding techniques from game theory, offering a computationally efficient strategy for optimizing the acquisition function, particularly GP-UCB.

**Strengths:**

- Large-scale combinatorial optimization is an important challenge, though Bayesian optimization (BO) may not be the most typical approach for tackling it.
- While theoretical analysis is provided for the convergence rate to an $\epsilon$-Nash equilibrium. However, it does not necessarily equate to finding the global optimum of the objective function, and the results are not significantly different from known results for existing game-theoretic BO literature.

**Weaknesses:**

- **Clarity**
The weakest part of this paper is definitely the clarity, particularly in math. There are so many uncommon, unexplained, and undefined symbols, that make me frustrated. Detailed below:

**Problem Statement**.  What exactly is $x$? I initially understood it as a $d \times n$-dimensional variable. However, in Line 107, the statement "f(x) corresponds to --- the amino acid sequence $x$, and each $x$ can take $20^n$". I've lost. Is $x$ meant to be a $d$-dimensional variable or a  $d \times n$-dimensional variable? I also can’t interpret the meaning of "each $x$ can take $20^n$". Shouldn't $d = 20$ in this case? Additionally, the definition of the fitness function $f(x)$ is unclear. Is $f(x)$ defined for each discrete variable, or is it applied to the entire variable set? For example, if $x$ is assumed to be $d$-dimensional, does our objective function take the form $\prod_{i=1}^n f(x^{(i)})$? Or is $x$ actually $d \times n$-dimensional, and thus $f(x)$ refers to something else? I’ll proceed under the assumption that $x$ is $d \times n$-dimensional.

**Section 3**. I understand the reward $r^{(i)}$ here is the UCB defined for $x$ as a $d \times n$-dimensional input, and each player $i$ corresponds to one of the discrete variables. However, since UCB is defined on the entire $d \times n$-dimensional space, how can we apply UCB as $r^{(i)}$ then? How do you select the other $n-1$ variables? Or does $r^{(i)}: \mathcal{X} \rightarrow \mathbb{R}$ imply that each player takes a $d \times n$-dimensional input and $r^{(i)}$ = UCB for all $i$? If that is the case, and all players share the same utility function, why do equilibria arise? Wouldn't this reduce to the same optimal points for all players? Additionally, in Definition 3.1, $r^{(i)}$ takes two arguments, but this is supposed to be UCB. Why does UCB take two arguments? What is arg eq? What exactly does Eq (2) represent? I'm also confused by the explanation on Lines 172-173: "Intuitively, equilibria are computed by breaking down the complex decision space into individual decision sets." Could you precisely explain this process before jumping into the intuition? How is finding a Nash equilibrium equivalent to maximizing UCB? What is the payoff $v$ in Algorithms 2 and 3? I've lost again.

I’ll stop pointing out specific issues here, but I want to note that the same level of confusion persists throughout the remaining sections (although the introduction was quite smooth).

- **Lack of Critical Guarantee**
Is finding a Nash equilibrium mathematically equivalent to maximizing the objective function $f$? If I understand correctly, this process is more akin to constructing a Pareto frontier, which contains the global maximum, making it more aligned with finding the set of local maxima. How, then, does this algorithm guarantee global convergence? If the goal of this work is local optimization or a heuristic approach, this should be clearly stated in the title or assumptions. Approximations can be valuable, but only if they provide some guarantee that they can recover the original problem under some conditions. Existing work, such as [1], provides such guarantees. I am not surprised that a local optimization algorithm finds local optima (as this work does) faster than a global optimizer (like [1]) in certain experiments, as they target different objectives. Even with these results, I wouldn’t choose this method for global optimization. When the function query is highly expensive, the computational overhead of the acquisition function becomes comparable. Moreover, [1] is not exactly state-of-the-art as it dates to 2022. There are likely more recent studies, such as [2], though performing a literature search is not my role here.
- **Limited Evaluation to Assess Practicality**
Heuristic approaches are completely acceptable if they prove useful in real-world applications. However, in that case, we expect the algorithm to be genuinely practical and perform best among existing heuristics. In the context of protein optimization, chemists would most likely start with deep learning-based approaches, especially diffusion-based methods, which are already well-established. There is a plethora of scalable and sample-efficient work in this area, such as [3, 4]. These methods seem like a more natural choice than using deep learning embeddings as features, as done in this work. Even with embeddings, the GP has no prior data on the objective function, meaning BO must start without a pretrained dataset. Alternatively, why not reduce the dimensionality of the embedding features to make them more tractable for the naïve GP-UCB? This could be achieved by adding just one additional layer to the transformer. In doing so, standard GP-UCB would likely perform just fine like popular latent BO doing. Since this paper does not compare its method against these popular alternatives or simpler alternatives, I cannot properly evaluate whether it represents a good heuristic. What I gather from this work is that it performs better than outdated methods on the limited tasks provided, but that alone does not demonstrate its superiority or practicality compared to modern approaches.
- **Limited Novelty**
The equation on line 295 appears nearly identical to the existing optimistic game-theoretic approach presented in [5], but it is not cited. Additionally, the idea of treating each discrete variable as a player is not particularly novel. For instance, Shapley value GP [6] and additive kernels [7] can be seen as variants of this concept (although they treat dimensions as players). While the specific combination of game theory, combinatorial optimization, and Bayesian optimization may be novel, it ultimately feels like a straightforward combination of existing approaches.
- **Missing Limitation**
I understand that this method leverages the problem structure where each dimension contains the same number of discrete variables, $n$. However, in typical combinatorial Bayesian Optimization, the goal is hyperparameter optimization, where each dimension may have a different number of categorical values, and the space can sometimes be a mixture of continuous and discrete variables. This method is not applicable to such cases. In mixed-variable scenarios, the existence of a Nash equilibrium cannot be guaranteed.

- **Citation**
- [1] Daulton, Samuel, et al. "Bayesian optimization over discrete and mixed spaces via probabilistic reparameterization."NeurIPS 2022
- [2] Papenmeier, Leonard, et al., "Bounce: reliable high-dimensional Bayesian optimization for combinatorial and mixed spaces." NeurIPS 2023
- [3] Gruver, Nate, et al. "Protein design with guided discrete diffusion." NeurIPS 2023
- [4] Campbell, Andrew, et al. "Generative flows on discrete state-spaces: Enabling multimodal flows with applications to protein co-design." arXiv preprint arXiv:2402.04997 (2024).
- [5] Han, Minbiao, et al., "No-Regret Learning of Nash Equilibrium for Black-Box Games via Gaussian Processes." UAI 2024.
- [6] Chau, Siu Lun, et al., "Explaining the uncertain: Stochastic shapley values for gaussian process models." NeurIPS 2023
- [7] Kandasamy, Kirthevasan, et al., "High dimensional Bayesian optimisation and bandits via additive models." ICML 2015.

**Questions:**

See the weakness section.

---

> ### Author Response · Authors · 2024-11-25
> **rebuttal**
>
> We thank the reviewer for their detailed feedback. We address your concerns and questions below.
>
> ---
> - **[Problem Statement Clarification]:** As we clearly state in line 103 that $x$ consists of $n$ discrete variables, hence $x=(x^1, \dots, x^n)$, which makes it $n$-dimensional. If each component of $x$, namely $x^i$ has $d$ possible values (or you can think of it as categories), then $x$ can take $d^n$ many configurations, that is the number of all possible values that $x$ can take. Within the protein design context, each discrete variable (protein site) $x^i$ can take $20$ possible (aminoacid) choices, hence, all possible values that the protein $x$ can take is $20^n$, where $n$ is the number of protein sites (discrete variables) and this makes $d=20$.
> Note line 102 that we already define $f$ over domain $\mathcal{X}$, which means over the possible values of $x$. Also note that $\mathcal{X}=\mathcal{X}^{(1)} \times \dots \times \mathcal{X}^{(n)}$. However, we cannot say that the objective function is of the form that the reviewer describes. Because $f$ is defined over the joint configuration $x$, not decomposable to individual discrete variables.
> ---
> - **[Clarification on the reward function]:** The reward $r^i$ is defined over the joint configuration $x$ and takes one argument which is $x$. We state, as the reviewer calls, two inputs “$x^i, x^{-i}$”, to show the joint configuration made by the strategy of player $i$ and the strategy of the rest of the players (denoted with $^{-i}$ notation). "arg eq" is an operator that returns the joint configuration(s) such that it forms an equilibrium according to UCB, hence equation (2) is a mathematical expression to show equilibria. The payoff function $v$ is the reward function in Definition 3.1. We have only used the conventional symbols from game theory in defining equilibrium finding subroutines. Finally, the equilibria are computed by breaking down the complex decision space into individual decision sets, because as we show in equilibrium finding subroutines (e.g. IBR), in each round the players select a strategy (action) by fixing the others’ strategies. Hence, this decomposes the global search over all possible configurations for $x$ to search over individual decision sets for $x^{(i)}$.
> ---
> - **[Guarantee]:** Finding the Nash equilibrium is not equivalent to maximizing the objective function $f$. We explained in Section 3 (line 181) that GameOpt leverages local optima. In Section 4, we provided a sample complexity analysis for GameOpt to reach an approximate equilibrium of the true objective function $f$. Section 4 clearly explains that GameOpt provides an approximation. We disagree with the statement that `“global optimization…the computational overhead of the acquisition function becomes comparable”`.  The global optimization (GP-UCB baseline) is intractable with $O(d^n)$ complexity, assuming that each dimension is of size $d$. Whereas, with e.g. GameOpt-IBR in which each player generates the best response w.r.t the previous joint strategy has the total complexity of $O(nd)$ assuming that each $\mid \mathcal{X}^{(i)} \mid = d$ for players $i=1,...,n$. Further, as we discuss the computational complexity in Appendix E4 and E3, our results show that in all domains, particularly for the larger spaces, GameOpt provides tractable acquisition function optimization with less compute amount required.
> ---
> - **[Novelty]:** As stated by the reviewer, the provided references operate on continuous spaces. Whereas, we clearly state in our paper that “novel game-theoretical approach **to combinatorial BO**”. As also stated by the reviewer `“While the specific combination of game theory, combinatorial optimization, and Bayesian optimization may be novel...”`, since there is no study that specifically combines game theory and combinatorial BO by treating each discrete variable as a player, we still think that this makes GameOpt novel **(also supported by the reviewers VxGg and mcjS)**.
> - - **[A clarification for the provided reference [5]]:** [5] seems to be published after July 2024, counted as a ‘very recent work’ by the ICLR guide (https://iclr.cc/Conferences/2025/ReviewerGuide) (see the last Q&A). We were unaware of this work, however, we are happy to cite it in our paper.
> - - While Shap and GameOpt both draw inspiration from game theory, their *applications and objectives* differ **significantly**. Shap is primarily used for *interpretability and explanation* by focusing on the contribution of individual dimensions to a prediction. In contrast, GameOpt leverages game theory to *optimize* a combinatorial space **by finding equilibrium points** of an acquisition function.
>
> ---
>
> > [5] Han, Minbiao, et al., "No-Regret Learning of Nash Equilibrium for Black-Box Games via Gaussian Processes." UAI 2024.

---

> ### Author Response · Authors · 2024-11-25
> **rebuttal-continue**
>
> - **[Additional Evaluation with Different Action/Dimension Sizes]:** GameOpt approach is **applicable** when each dimension $x^i$ (players) has a different action set ($\mathcal{X}^{(i)}$) size. We provide additional experiments in Appendix E12 considering different action set sizes. The results in Figure 17 clearly demonstrate the **applicability and generalizability** of GameOpt to such problem domains.  Further, there is no significant performance difference between individual players and player grouping settings with varying action set sizes. We also modified our statement (lines 104-105) in the main text.
> ---
> - **[Comparison with not relevant protein optimization methods]:** As baselines, we considered the combinatorial BO methods that **target efficient acquisition function optimization**. One can find a vast amount of diverse approaches (from generative models to LLMs, etc.) to apply to the protein design problem. However, in our study, we propose GameOpt for combinatorial BO, with an application to an interesting real-world problem (which demonstrates its success and applicability in such a complex problem). Hence, comparisons against all other protein design approaches are beyond our scope.
> - - `“...why not reduce the dimensionality of the embedding features to make them more tractable for the naïve GP-UCB?”`:  In our paper, we address acquisition function optimization over large combinatorial search spaces. Even though we’d reduced the dimensionality of the embedding features for the GP surrogate, that would not change the **searching over combinatorial space** to sample evaluation points by optimizing the acquisition function. Hence, that would only help for the surrogate model fitting part, but **not the main problem addressed in our study**. We hope this clarifies the misunderstanding. We also want to highlight that the proposal to reduce dimensionality is not as trivial as the reviewer makes it appear. We need to explore regions where potentially no data is available and dimensionality reduction methods would most likely ignore this altogether.
> - - **[Outdated methods]:** We identify appropriate baselines that target acquisition function optimization, as this is our main focused problem. Only to provide interesting insight, we further experiment on benchmark 25D-PestControl problem [1]. As given in Appendix-E13 of the updated manuscript, we compare our GameOpt approach against some high dimensional combinatorial BO methods [2,3], showing (see Figure 19) that GameOpt provides faster convergence to better solutions compared to these methods.
> - - **[Additional task]:** We performed experiments on domains with different combinatorial search space sizes, from 20^3 to 20^55. Since the main problem we address is how to **efficiently** and strategically optimize the acquisition function over large combinatorial search spaces, we think that demonstrating the performance over a diverse size of search space is aligned with our addressed problem, and supports our empirical evaluation.
> However, to address your concern, we further experimented on the benchmark 25D-PestControl problem [1], with search space size 5^25, as explained above.
>
> ---
> Overall, with the additional evaluations we provided, we showed the applicability of GameOpt under various problem domains with different tasks and domain structures, and against diverse methods. We hope that these points clarified your concerns. We are happy to expand them further if needed.
>
> ---
>
> > [1] Changyong Oh, Jakub Tomczak, Efstratios Gavves, and Max Welling. Combinatorial bayesian optimization using the graph cartesian product. Advances in Neural Information Processing Systems, 32, 2019.
>
> > [2] Leonard Papenmeier, Luigi Nardi, and Matthias Poloczek. Bounce: Reliable high-dimensional bayesian optimization for combinatorial and mixed spaces. Advances in Neural Information Processing Systems, 36:1764–1793, 2023.
>
> > [3] Aryan Deshwal, Sebastian Ament, Maximilian Balandat, Eytan Bakshy, Janardhan Rao Doppa, and David Eriksson. Bayesian optimization over high-dimensional combinatorial spaces via dictionary-based embeddings. In International Conference on Artificial Intelligence and Statistics, pp. 7021–7039. PMLR, 2023.

---

> ### Author Response · Authors · 2024-11-27
>
> Dear Reviewer XU81,
>
> As we approach the end of the discussion period, we would like to ask if there is any additional information we can provide that might influence your evaluation of the paper. We believe our previous response with extensive explanations and additional experiments conducted has effectively addressed your initial concerns and clarified some points that may have been overlooked.
>
> Thank you once again for your consideration of our work.

---

> > ### Comment · Reviewer_XU81 · 2024-11-27
> >
> > Thank you for your detailed response. While your rebuttal has addressed some of the minor points I raised, the major issues remain unresolved. As a result, I will maintain my score and continue to recommend a major revision based on the following concerns:
> >
> > ## 1. Clarity
> > Although your explanation is appreciated, the main text of the draft has not been updated to reflect the clarifications provided in the rebuttal. This means that potential readers are likely to encounter the same confusion I highlighted in my previous comments. Consequently, the clarity of the paper remains insufficient. As stated earlier, a major revision is necessary to resolve this issue. I will refrain from reiterating the specific points I previously raised.
> >
> > ## 2. Weak Theoretical Contributions
> > The theoretical contributions of the paper are not sufficiently robust. While the work focuses on local optimization, it does not address global optimization, which is the standard focus of most theoretical Bayesian optimization (BO) papers. For example, BO papers typically provide regret bounds for global optimizers, a standard that is absent here. As such, the theoretical contributions alone are not strong enough to support acceptance.
> >
> > ## 3. Weak Empirical Evaluations
> > It seems my previous comments may not have been fully understood, so I will rephrase them for clarity. The main issue with the empirical evaluations lies in the lack of alignment with the expectations of the paper’s intended audience. Based on the topic, the primary audience for this paper could be categorized into two groups:
> > - (a) Experimental BO researchers: For this audience, strong empirical justification is critical. They would expect the proposed method to outperform existing algorithms on widely recognized benchmark test functions. This is essential because, as the no free lunch theorem suggests, no algorithm can universally outperform others. Therefore, demonstrating the method’s benefits across multiple standard test functions is necessary. However, this paper evaluates the method only on a new dataset, making it difficult to compare its performance to existing algorithms.
> > - (b) Protein design researchers: This audience would expect a practical algorithm that provides a significant advantage for protein design tasks. For them, the baseline comparisons should extend beyond BO algorithms to include widely accepted approaches in the protein design field, such as diffusion-based models. Moreover, the paper does not justify why BO is particularly suitable for protein design over these alternatives.
> >
> > In summary, the empirical results fail to convincingly address the expectations of any of these three key audiences: BO theoreticians, experimental BO researchers, or protein design researchers. As a result, the paper feels unfocused, and its novelty justifications remain weak.
> >
> > ## Conclusion
> > To address these concerns, I recommend a major revision that improves clarity, strengthens the theoretical contributions, and provides more compelling empirical evaluations tailored to the expectations of the paper’s intended audience. Additionally, refining the text to better focus on the target audience and clearly convey the novelty and significance of the work is essential.

---

> ### Author Response · Authors · 2024-11-27
> **further clarification**
>
> We thank the reviewer for the follow-up comments, which we clarify them below:
>
> ---
>
> **1. Clear notation is provided in the main text**
>
> - We have re-iterated our rebuttal (which we provided to you) and made the necessary changes in the main text (see updated manuscript). For the payoff/reward function notation, we have updated our manuscript accordingly. However, we identified minor changes as we used the same notation and statements both in our explanation for rebuttal and the manuscript.
>
> - Also note that all other reviewers found the presentation clear with a good score, stating that `"The presentation is clear and the idea is easy-to-follow."`. Hence, instead of a major revision, the minor editions we made in the updated manuscript have improved the clarity of our text.
>
> ---
>
> **2. Our claim is theoretically backed up for local optimality & a discussion on the global optimality challenge is already provided in the main text**
>
> - As noted by the reviewer, while presenting GameOpt, we state it as "A form of local optimality", given the discussion paragraph in Section 3. For a theoretical backup for this, we already provided a sample complexity analysis (Section 4).
>
> - We already carefully addressed about global optimality with the "price of anarchy (PoA)" discussion in Section 3. We have explained how such PoA guarantees do not readily apply to our setting because of a lack of submodularity of the objective function. And theoretically bounding the suboptimality and steering players to the best equilibrium is an active and challenging area of research. We already provided a detailed discussion to justify why evaluating GameOpt solely as an optimization technique is challenging, and how GameOpt’s filtering-out suboptimal equilibria step contributes to its quality and robustness.
>
> ---
>
> **3. Comprehensive experiments with ablation studies**
>
> - We propose GameOpt **for combinatorial Bayesian optimization** with an application to protein design. As we stated in our paper (lines 17-20,  68-70, 82-84, etc), our main problem focus is **tractable acquisition function optimization in BO over large combinatorial spaces.**
> - Considering this, we identified our baselines from related works on combinatorial BO which directly target acquisition function optimization. We have presented 6 main problem domains on which we evaluate our proposed method fairly against baselines under **various combinatorial search space sizes**. We consider **real-world datasets** and problem context, which further shows the applicability of our method on real-world important problems. We have provided many ablations, including computational complexity, batch size analyses and further comparisons to provide interesting insight.
> - As also stated by Reviewer 3y4p (`”It’s good to see substantial experiments…Experiments are comprehensive and ablation studies are provided”`), our work includes extensive analysis with detailed ablations, providing in-depth insights into the performance and flexibility of our method.
>
> ---
>
> We hope in the light of above clarifications, you could reconsider your evaluation.

---

### Official Review · Reviewer_mcjS · 2024-11-02

**Soundness:** 2
**Presentation:** 3
**Contribution:** 2
**Rating:** 3
**Confidence:** 3

**Summary:**

This paper studied the combinatorial Bayesian optimization in the discrete search space. Based on the theory of cooperative games and upper confidence bound of Gaussian processes (GPs), a new acquisition function that seeks to find the local equilibria of players was proposed. Based on batch and parallel computation, the proposed method achieved better scalability. The effectiveness and competitiveness were demonstrated in the application of computational protein design.

**Strengths:**

1. The discrete black-box optimization problems are important, and the proposed game-theoretical solution seems novel.
2. The protein design is vital in drug discovery and healthcare.
3. The presentation is clear and the idea is easy-to-follow.

**Weaknesses:**

1. There is no mechanism to avoid or mitigate stucking in local optimality. At the same time, as a Bayesian optimization method, the exploration-exploitation balance is not considered. As a preliminary solution, random restarts can be introduced into the search process, similar to what the trust-region Bayesian optimization algorithm does [1], to ensure exploration and global search ability.
2. As local optima, the regret bound is not insightful enough to show the superiority. Currently, the regret is similar to what the original GP-UCB is, but only local optima are guarenteed. While the GP-UCB ensures global search and convergence. In my opinion, assuming convexity in the local region containing equilibria may help improve the theoretical results [2].
3. The testing benchmark problems are not diverse enough. Some well-known synthetic problems are required for fair comparison, espectially that employed in the state-of-the-art work.
4. The influence of hyper-parameters, especially the batch size B, is not clear. The baseline algorithms, especially the GP-UCB, are not tailored for batch optimization. It is recommended to consider smaller B, such as B=1, first.
5. The literature review is insufficient. More baseline algorithms in combinatorial Bayesian optimization and general optimizers should be considered.

[1] David Eriksson, et al.: Scalable Global Optimization via Local Bayesian Optimization. NeurIPS 2019.

[2] Shuang Li, et al.: "Why Not Looking backward?" A Robust Two-Step Method to Automatically Terminate Bayesian Optimization. NeurIPS 2023.

**Questions:**

1. Is there any consideration of balance of exploration and exploitation? How is the data-efficiency guaranteed?
2. What is the influence of batch size B? In what case may the algorithm perform worse?
3. Why not compared baselines using tree-based surrogate models. As far as I know, the tree model is a suitable choice for protein design.
4. How to prove that solving equation (2) is more tractable than solving other acquisition functions?

---

> ### Author Response · Authors · 2024-11-25
> **rebuttal**
>
> We thank the reviewer for their feedback. We are happy that you recognize the novelty of our GameOpt framework, with clear idea and presentation on an important real-world problem application. We address your concerns and questions below, including additional experiments and paper edits inspired by your comments.
>
> ---
> - **[GameOpt balances exploration and exploitation & Q1]:** In GameOpt, we indeed **balance exploration and exploitation**. Mainly, the **UCB** itself captures the exploration-exploitation tradeoff. GameOpt further has the following mechanisms, which help to address this:
> - - (1): We introduce random restarts in the M equilibrium computation step. As given in Algorithm-4 in Appendix B, the M parallel games with random initializations collect M equilibria. As given in step 5, we start with a random protein sequence. As also stated in Algorithms 2 and 3 (step 2), each of these games starts with a random joint strategy.
> - - (2): We implement filtering (see step 5 of Algorithm-1). We collect a larger amount of M equilibria, then select the top-B.
> We hope that this also clarifies your first question.
> ---
> - **[Theoretical results]:** We thank the reviewer for their suggestion on assuming convexity. However, note that [1] operates on a continuous domain. Whereas, we are not sure how convexity directly can be applied to our combinatorial setting. Submodularity could have been leveraged, only if, the objective function were submodular. And this does not readily apply to our setting. The regret bound can be similar to GP-UCB, however, as we extensively demonstrate with our empirical evaluations GameOpt achieves a similar regret level with a **tractable** approach, which is important as the combinatorial search space scales up.
> ---
> - **[Additional problem domain]:** We experimented on problem domains with **different sizes of combinatorial search spaces**, from 20^3 to 20^55. Since the main problem we address is how to **efficiently** optimize the acquisition function over large combinatorial search spaces, demonstrating the performance **over a diverse size of search space** is already aligned with our addressed problem.
> - - However, to address your concern, we further experiment on the benchmark 25D-PestControl problem [2], with a search space size 5^25. As given in Appendix-E13 of the updated manuscript, we compare our GameOpt approach against some high dimensional combinatorial BO methods [3,4]. The additional analysis clearly supports (see Figure 19) that GameOpt **generalizes** well to other problem domains, providing an outperforming performance.
> ---
> - **[Batch size more ablations & Q2]:** We now also show the performance comparison under batch sizes $B=1$ and $B=3$ on Halogenase and GB1(4) domains, in Appendix-E7 of the updated manuscript. As given in Figure-12, even under a more restricted setting on both problem domains, GameOpt achieves superior performance against baselines, although initially surpassed by GP-UCB under batch size 1.  As the batch size increases, GameOpt’s optimistic game approach benefits from parallelism and collects diverse local optima, hence, GameOpt surpasses. We hope that this also clarifies your second question.
> ---
> - **[Related Work]:** We would like to clarify that the provided references operate on continuous domains, whereas we focus on combinatorial domains. We nevertheless included the provided works in our related work section of the updated manuscript. Furthermore, we considered combinatorial BO approaches that are mainly categorized into two: (1) targeting the acquisition function optimization–which is our main focus, and (2) targeting the surrogate modeling with discrete variables. We added the provided references to the second category, for which we offer a general overview, as our primary emphasis remains on the first category.

---

> ### Author Response · Authors · 2024-11-25
> **rebuttal-continue**
>
> - **[Surrogate model choice & Q3]:** For a fair comparison, we selected methods that target acquisition function optimization over large combinatorial domains (our main focus). Using some other type of surrogate modeling would not be fair as in our evaluation we **fix the surrogate model** for all BO methods and evaluate different acquisition function optimization methods. Please also see that in Appendix-E11 (Figure 16) we already provide additional experiments using surrogate models with different design choices. Specifically, we use GP defined with string kernels, which clearly demonstrates that GameOpt is effective under different surrogate model design choices.
> ---
> - **[Tractability of GameOpt]:** With equation (2), we decouple the combinatorial search space into individual search (decision) sets (as illustrated in Figure-1). For instance, considering GameOpt-IBR, each player generates the best response w.r.t the previous joint strategy with the total complexity of $O(nd)$ assuming that each $\mid \mathcal{X}^{(i)} \mid = d$ for players $i=1,...,n$. Step 5 of Algorithm 2 finds the maximum valued element in a set with fixed size “n”, that is the number of players. Note that the global optimization which is performed by GP-UCB is intractable with $O(d^n)$ complexity. We provided a detailed discussion on the complexity and computational cost in Appendix E4 and E3.
> ---
> Overall, with the additional evaluations we provided, we showed the applicability of GameOpt under various problem domains with different tasks and domain structures, and against diverse methods. We hope that these points clarify your concerns. We are happy to expand them further if needed.
>
> ---
>
> > [1] Shuang Li, et al.: "Why Not Looking backward?" A Robust Two-Step Method to Automatically Terminate Bayesian Optimization. NeurIPS 2023.
>
> > [2] Changyong Oh, Jakub Tomczak, Efstratios Gavves, and Max Welling. Combinatorial bayesian optimization using the graph cartesian product. Advances in Neural Information Processing Systems, 32, 2019.
>
> > [3] Leonard Papenmeier, Luigi Nardi, and Matthias Poloczek. Bounce: Reliable high-dimensional bayesian optimization for combinatorial and mixed spaces. Advances in Neural Information Processing Systems, 36:1764–1793, 2023.
>
> > [4] Aryan Deshwal, Sebastian Ament, Maximilian Balandat, Eytan Bakshy, Janardhan Rao Doppa, and David Eriksson. Bayesian optimization over high-dimensional combinatorial spaces via dictionary-based embeddings. In International Conference on Artificial Intelligence and Statistics, pp. 7021–7039. PMLR, 2023.

---

> > ### Comment · Reviewer_mcjS · 2024-11-27
> > **Response to the Authors' feedback**
> >
> > I believe a good empirical design can be appreciated without insightful theoretical analysis, as long as the experimental study is extensive. However, in terms of the experiments, my concerns exist.
> > 1. The baseline algorithms. The authors categoried the combinatorial BO methods into surrogate design and acquisition design, and choose not to compare their method with ones using parametric models as reviewed in Lines 239-246. However, the methods in the context of acquisition design are rather limited, as given in Line 247. Since all methods follow the combinatorial BO routine, I do not understand why the authors choose not to compare their method with others, even if they relied on domain-specific design for better performance.
> > 2. The inconsistency. Based on the above problem, why were there only the additional baseline algorithms considered in Section E.13? I am not convinced that Bounce and BODi cannot be used in protein design (even their performance may not be good enough). Likewise, why were GP-UCB, IBR-FITNESS and PR not considered in Section E.13?
> > 3. The superiority. Does GameOPT significantly outperform other method in statistics? Quantitive study such as the Wilcoxon rank-sum test is highly recommended.

---

> ### Author Response · Authors · 2024-11-27
> **Further clarification**
>
> We thank the reviewer for the follow-up comments, which we clarify them below:
>
> ---
>
> - **[Appropriate baselines & fair comparison]:** We highlight that our problem focus is: **tractable acquisition function optimization in BO over large combinatorial spaces.**
> - 1. For a fair and meaningful comparison and evaluation of our proposed method, we considered “baselines” from the **same problem focus**. The mentioned methods (e.g. Bounce, BODi) are not directly comparable, because they address creating nested embeddings for the GP surrogate model. In our method, we **fix the GP surrogate** for the BO methods and compare the acquisition function optimizers. Hence, comparing against those methods directly would not provide a meaningful evaluation because **they don’t target the same problem**. We can only integrate our method into their framework’s acquisition function optimization, which we already did and showed in additional experiments presented in Appendix E13.
> - 2. Bounce and BODi use local search to optimize the acquisition function, which we have already compared our GameOpt approach against in the protein design context.
> - 3. As also stated by Reviewer 3y4p (`”It’s good to see substantial experiments…Experiments are comprehensive and ablation studies are provided”`), our work includes extensive analysis with detailed ablations, providing in-depth insights into the performance and flexibility of our method.
> - 4. From an **optimistic perspective**, the relatively limited number of methods in the context of acquisition optimization highlights how *underexplored* this specific problem is. This further underscores the **relevance and contribution of GameOpt** in this context.
> ---
> - **[Additional analysis in Appendix E13]:** As we explain above, additional methods are not GameOpt's baselines. To clarify, we only provide further analysis in Appendix E13 to show how GameOpt can also be **integrated** with other high-dimensional BO methods (to their acquisition function optimization). This analysis was already beyond our scope, nevertheless, supported GameOpt’s effectiveness even more & showed interesting insight & addressed your concern.
>
> - 1. GP-UCB cannot be added because it is **intractable**, which we try to address this problem throughout our paper.
> The problem search space is 5^25 (much larger than the search spaces we had shown in GP-UCB results), and it is not computationally feasible to search for the optimal solution within this huge space in each BO iteration.  Also note that, none of the high-dimensional BO methods that the reviewer is most concerned about, compare their methods against GP-UCB.
> - 2. Showing how well the PR method integrates with other high-dimensional BO methods is not in our scope. Hence it is not included in Appendix E13. We have already shown with extensive experiments on various search space sizes and batch size settings that GameOpt outperforms PR by discovering better solutions much faster.
> - 3. Lastly, Bounce and BODi use local search to optimize the acquisition function, which we have already compared our GameOpt against such local search methods in the protein design context, as presented in Appendix E3.
> ---
> - We appreciate the statistical test suggestion, however, our results clearly demonstrate that GameOpt outperforms baselines. It **converges much faster, to higher fitness-valued solutions**. Further, how we evaluate the performance is already aligned with the considered related works. Note that, all previous works that the reviewer suggested, measure the performance by tracking the metric: **best objective value observed throughout BO iterations**. We, on the other hand, also provide additional metrics detailed in Section 5 and with further analyses in Appendix E.
> ---
>
> We hope that these key points clarified your questions.

---

### Official Review · Reviewer_3y4p · 2024-11-03

**Soundness:** 3
**Presentation:** 3
**Contribution:** 4
**Rating:** 8
**Confidence:** 4

**Summary:**

This paper studies how to solve the combinatorial Bayesian optimization problem, which has many important potential applications like protein design. The key idea of this work is introducing the Nash equilibrium to Bayesian optimization, where a game is carefully designed for domain variables to play and the decision making is driven by equilibrium finding algorithms. It’s good to see substantial experiments in protein design in the end.

**Strengths:**

1.	Combinatorial Bayesian optimization (CBO) is a very important subfield in Bayesian optimization. While Bayesian optimization has been successfully applied to many areas, studying CBO helps us solve even more practical application problems, like protein design.
2.	Sample complexity guarantees are provided to show the algorithm’s ability to achieve approximate Nash equilibrium.
3.	Experiments on protein design are comprehensive and ablation studies are provided.

**Weaknesses:**

1.	To be honest, as a ML researcher, I have limited background in Nash equilibrium and I think it could be beneficial for this paper to provide more information on introducing Nash equilibrium since this is an ICLR submission. Although locally optimal, why and how does Nah equilibrium help in CBO setting? Could you provide a brief comparison between the local optimality of Nash equilibria and global optimality in the context of CBO. Can we design some globally optimal CBO algorithms or are the authors aware of such algorithms? Also, it might be hard to theoretically investigate the performance difference between GameOPT and the global optimal solution, it is feasible to investigate it in experiments where a small finite decision space is used. That would greatly provide valuable insights into the method's effectiveness.
2.	I have concerns on the computational efficiency of Step 5 of Algorithm 1. Because the GP is assumed on function f which is now defined on the combinatorial space, how do you find the top B equilibria according to UCB? Could you provide more details on how to implement this step efficiently, discuss the computational complexity of this step compared to the overall algorithm, and explain any approximations or heuristics that may be used if an exact solution is computationally infeasible?

**Questions:**

1.	Does this algorithm work without assumption $|\mathcal{X}^{(i)}|=d$? It means that in this case the numbers of possible decision variables are different.
2.	What is the v function in Algorithm 2? Is that the r function defined in Definition 3.1?

---

> ### Author Response · Authors · 2024-11-25
> **rebuttal**
>
> We thank the reviewer for their feedback and for supporting our paper! We are happy that you recognize the sample efficiency guarantee for the proposed GameOpt framework and our comprehensive empirical evaluation with ablation studies. We address your concerns and questions below.
>
> ---
>
> - **[Nash equilibrium insight]:** (1) Our sample complexity analysis provides an insight into the solution obtained by the GameOpt and how it guarantees an approximation ($\epsilon$-approximate Nash equilibrium) of the underlying function $f$. (2) We have already shown with GB1(4) experiment results (see Figure-2(b)) how close the solutions obtained via GameOpt-IBR and GameOpt-Hedge to the “Best Solution”, that is the global optimal solution for the GB1(4) problem. Further, as shown in Appendix-E7 (Figure 11), when batch size is increased, GameOpt benefits from collecting more local optima and converges much faster to the global optimal solution compared to other baselines.
> ---
> - **[Computational efficiency of GameOpt - Step 5]:** The steps 4 & 5 are the key steps that make GameOpt tractable compared to global optimizer (GP-UCB). For instance, considering GameOpt-IBR, the total complexity of generating equilibria is $O(nd)$ assuming that each $\mid \mathcal{X}^{(i)} \mid = d$ for players $i=1,...,n$. Note that the global optimization which is performed by GP-UCB is intractable with $O(d^n)$ complexity.
> ---
> - **[Q1 & Additional experiments with varying dimension (action set) sizes]:** The GameOpt approach is applicable when each dimension $x^{(i)}$ (players) has a different action set ($\mathcal{X}^{(i)}$ ) size. Prompted by your feedback, we provide additional experiments in Appendix E12 considering different action set sizes. The results in Figure 17 clearly demonstrate the **applicability** and **generalizability** of GameOpt to problem domains with unequal action set sizes.  Further, there is no significant performance difference between individual players and player grouping settings with varying action set sizes. To clarify this, we also modified our statement (lines 104-105) in the main text.
> ---
> - **[Reward function $v$]:** Yes, the payoff function $v$ is the reward function $r$ in Definition 3.1. We have only used the conventional symbols from game theory in defining equilibrium finding subroutines.
> ---
> Overall, with the additional evaluations we provided, we showed the applicability of GameOpt under various problem domains with different tasks and domain structures, and against diverse methods. We hope that these points clarify your concerns. We are happy to expand them further if needed.

---

### Official Review · Reviewer_VxGg · 2024-11-06

**Soundness:** 3
**Presentation:** 3
**Contribution:** 3
**Rating:** 8
**Confidence:** 3

**Summary:**

The paper proposes a novel method that frames the protein design task as an optimization problem of finding a Nash Equilibrium (NE) with unknown utility functions. It provides sample efficiency guarantees, and empirical results using multiple NE solvers suggest promising improvements over existing baselines.

**Strengths:**

- ***Innovative Formulation***: The paper introduces an interesting formulation by shifting the focus from kernel design for protein structures to an objective that reflects the insight that multiple separate contributing factors exist within protein design tasks.

- ***Sample Efficiency Guarantee***: The algorithm comes with a sample efficiency guarantee, which is valuable for practical scientific experimental design tasks where data collection can be costly and demand a principled optimization solution.

**Weaknesses:**

- ***Limited Novelty in BO Guarantees***: The Bayesian Optimization (BO) component naturally inherits the guarantees of the Upper Confidence Bound (UCB) algorithm. There doesn't appear to be a significant difference or improvement, especially considering that the method does not address specific challenges unique to protein design.
- ***Lack of Comparison with High-Dimensional BO Methods***: BO methods tackling protein design typically involve specific treatments for high-dimensional spaces. Including comparisons against these methods would strengthen the paper, especially when ESM embedding allows the direct application of such methods.

**Questions:**

- ***Challenges in Finding Nash Equilibrium***: Given that finding a Nash Equilibrium is PPAD-complete and challenging even with state-of-the-art solvers, could the authors provide more insight and detailed discussion on optimizing the equilibrium for protein applications, especially in high-dimensional cases? The paper could better illustrate why the benefits of introducing the equilibrium outweigh the additional challenges rather than implying that solvers are guaranteed to find the NE. This clarification would prevent confusion among readers not familiar with the literature, especially since multiple NE solvers are discussed while claiming guarantees of finding local optima.

- ***Price of Anarchy Discussion***: The discussion of the price of anarchy briefly addresses the motivation for the problem formulation but is not sufficiently convincing. Providing more details and insights specific to protein design tasks could improve the presentation and strengthen the argument.

p.s. Due to my limited domain expertise, I am not confident in assessing the novelty of the problem formulation in this specific application.

---

> ### Author Response · Authors · 2024-11-25
> **rebuttal**
>
> We thank the reviewer for their feedback. We are happy that you recognize our GameOpt framework as interesting, and acknowledge its sample efficiency guarantee. We address your concerns and questions below.
>
> ---
>
> - **[BO guarantees]:** The regret bound can be similar to the global optimizer GP-UCB, however, as we extensively demonstrate with our empirical evaluations that GameOpt achieves a similar regret level with a **tractable** approach compared to intractable GP-UCB, which is crucial as the combinatorial search space scales up.
> ---
> - **[Comparison with high-dimensional BO methods]:** As we discussed in Section 3.2, the combinatorial BO methods can be categorized into two: (1) the methods directly focus on surrogate modeling with discrete variables, and (2) the methods addressing acquisition function optimization within discrete search spaces. Since our proposed approach GameOpt falls into the second category, we identified our baselines from those methods. Hence, in our evaluation, we fix the surrogate model the same for all BO methods and evaluate different acquisition function optimization methods, unlike other high-dimensional BO methods that operate on surrogate modeling. Hence, such suggested methods would not be appropriate “baselines” for GameOpt.
> - - However, to address your concern, we further compare GameOpt against two high-dimensional BO methods [1,2] on a benchmark 25D-PestControl problem [3], with a search space size 5^25. We provide this further analysis in **Appendix E13** of the updated manuscript. This additional experiment clearly shows (see Figure 19) that GameOpt provides faster convergence to better solutions compared to these methods, supporting its integration with other high-dimensional methods. Further, it shows that GameOpt generalizes well to other combinatorial BO problem domains.
> ---
> - **[Finding Nash equilibrium]:** As we discuss in Section 3.1. (lines 234-235), we do not imply that solvers are always guaranteed to find the NE; This is true only under some conditions (line 236). In fact, we introduce Algorithm-3 by saying that it finds coarse correlated equilibrium only.
> ---
> - **[Price of Anarchy]:** As we stated in Section 3 (lines 196-197), the provided PoA guarantee does not readily apply to our setting, because the underlying objective function (protein fitness function in the context of protein design) is not submodular. Further, bounding the suboptimality and steering players to the best equilibrium is an active and challenging area of research. We already provided a detailed discussion to justify why evaluating GameOpt solely as an optimization technique is challenging, and how GameOpt’s filtering-out suboptimal equilibria step contributes to its quality and robustness (lines 200-202).
>
> ---
> Overall, with the additional evaluations we provided, we showed the applicability of GameOpt under various problem domains with different tasks and domain structures, and against diverse methods. We hope that these points clarify your concerns. We are happy to expand them further if needed.
>
> ---
> > [1] Changyong Oh, Jakub Tomczak, Efstratios Gavves, and Max Welling. Combinatorial bayesian optimization using the graph cartesian product. Advances in Neural Information Processing Systems, 32, 2019.
>
> > [2] Leonard Papenmeier, Luigi Nardi, and Matthias Poloczek. Bounce: Reliable high-dimensional bayesian optimization for combinatorial and mixed spaces. Advances in Neural Information Processing Systems, 36:1764–1793, 2023.
>
> > [3] Aryan Deshwal, Sebastian Ament, Maximilian Balandat, Eytan Bakshy, Janardhan Rao Doppa, and David Eriksson. Bayesian optimization over high-dimensional combinatorial spaces via dictionary-based embeddings. In International Conference on Artificial Intelligence and Statistics, pp. 7021–7039. PMLR, 2023.

---

> ### Author Response · Authors · 2024-11-27
>
> Dear Reviewer VxGg,
>
> As we approach the end of the discussion period, we would like to ask if there is any additional information we can provide that might influence your evaluation of the paper. We believe our previous response with additional experiments conducted has effectively addressed your initial concerns and clarified some points that may have been overlooked.
>
> Thank you once again for your thoughtful consideration of our work.

---

> > ### Comment · Reviewer_VxGg · 2024-11-28
> >
> > I appreciate the authors' rebuttal and the effort to address the concerns raised. However, I still have the following questions and comments:
> >
> > 1. **Additional Baselines**: I share mcjS's concern regarding the claim that the suggested methods would not be appropriate baselines for GameOpt. Given the use of the ESM-1v embedding, I don’t see a significant limitation in applying methods like TuRBO, its variants, random embedding approaches, or variable selection methods designed for discrete search spaces. For instance, the following work demonstrates the applicability of such methods in related contexts:
> >
> >    ***Zhang, Fengxue, et al. "Learning regions of interest for Bayesian optimization with adaptive level-set estimation." In International Conference on Machine Learning, pp. 41579-41595. PMLR, 2023.***
> >
> >    While I do not expect an exhaustive empirical comparison, I believe a more comprehensive discussion of related works could enhance the soundness of the study.
> >
> > 2. **Nash-Equilibrium Challenges**: The rebuttal raised additional questions about the proposed problem formulation. Specifically, given the potential downsides of introducing a Nash-Equilibrium (NE) subroutine—such as the risk of overlooking the global optimizer and the inherent complexity of finding NE—I'm curious how the proposed approach can generalize effectively to broader benchmarks. Could the authors elaborate further on how these challenges are outweighed by the potential benefits?

---

> > > ### Author Response · Authors · 2024-12-03
> > > **further clarification**
> > >
> > > We thank the reviewer for the follow-up comments, which we clarify them below:
> > >
> > > ---
> > > 1. **Additional Baselines:** The use of ESM-1v embedding is for the surrogate modeling, which we **fix for all BO-based methods** to fairly compare and evaluate our game theoretic acquisition function optimization against other acquisition function optimizers on combinatorial search spaces. Hence, those are not suitable to be baselines, as we do not target surrogate modeling with different embeddings, but rather efficient acquisition function optimization over combinatorial spaces.
> > > - - TurBO, the provided reference and variants operate on continuous domains, they build trust region on latent space. Whereas GameOpt directly operates on combinatorial search spaces. Although not relevant, we have already shown a comparison against latent space optimizers in Appendix E11. Furthermore, we have already compared our method against local search methods by mimicking the trust region concept with fixed neighborhood size, presented in Appendix E3.
> > > - - We are happy to add the reference to our related work, in which we have provided a discussion on how these methods target different problem than ours.
> > > ---
> > > 2. **Nash-Equilibrium finding:** The key potential benefit is that GameOpt is **tractable** compared to the global optimizer (GP-UCB). For instance, considering GameOpt-IBR, the total complexity of generating equilibria is $O(nd)$ assuming that each $\mid \mathcal{X}^{(i)} \mid = d$ for players $i=1,...,n$. Note that the global optimization which is performed by GP-UCB is intractable with $O(d^n)$ complexity.
> > > - - (1) Our sample complexity analysis provides an insight into the solution obtained by the GameOpt and how it guarantees an approximation ($\epsilon$-approximate Nash equilibrium) of the underlying true function. (2) We have already shown with GB1(4) experiment results (see Figure-2(b)) how close the solutions obtained via GameOpt-IBR and GameOpt-Hedge to the “Best Solution”, that is the global optimal solution for the GB1(4) problem. Further, as shown in Appendix-E7 (Figure 11), when batch size is increased, GameOpt benefits from collecting more local optima and converges much faster to the global optimal solution compared to other baselines.
> > > ---
> > > We hope that these key points clarified your questions.

---

> > > > ### Comment · Reviewer_VxGg · 2024-12-03
> > > >
> > > > I appreciate the authors' responses. Their argument differentiating between model learning and acquisition optimization makes sense, and the elaboration on the NE finding clarifies my question. I will slightly increase my score.

---

> > > > > ### Author Response · Authors · 2024-12-03
> > > > >
> > > > > Thank you so much for raising your score and supporting our paper! We are happy to hear that the reviewer appreciates our clarifications. Your feedback has helped us improve our paper, and we sincerely appreciate that.

---

### Author Response · Authors · 2024-11-25
**General Response**

We appreciate the reviewers' time, positive feedback, and constructive comments.

The reviewers highlighted the novelty of our approach (`“proposed game-theoretical solution seems novel”`and `“introduces an interesting formulation”` (VxGg, mcjS, XU81)), its computational efficiency (`“offering a computationally efficient strategy”` (XU81)), and the clarity of its presentation (`“presentation is clear and the idea is easy to follow”` (mcjS)). They also noted the sample efficiency analysis (`“algorithm comes with a sample efficiency guarantee”` (VxGg, 3y4p)) and extensive evaluations with ablation studies (`“comprehensive and ablation studies”` (3y4p)) on a critical real-world domain (`“protein design is vital”` (mcjS)).

In summary, our key contribution is GameOpt: a novel game-theoretical approach to combinatorial Bayesian optimization, enabling efficient and tractable acquisition function optimization in large combinatorial domains.

---
Considering the reviewer’s comments, we updated our manuscript (modified parts **highlighted in “red”**), with the following additions:
- **[Additional experiments with varying action set (dimension) sizes]:** GameOpt approach is applicable when each dimension $x^{(i)}$ (players) has  different action set ($\mathcal{X}^{(i)}$) sizes. To demonstrate this, we provide additional experiments in Appendix E12 on two problem domains. The results in Figure 17 clearly demonstrate the applicability and generalizability of GameOpt to problem domains with unequal action set (dimension) sizes.  We also modified our statement (line 104-105) in the main text to clarify this.

- **[Comparison with high-dimensional BO methods]:**  We further compare GameOpt against two high-dimensional BO methods [1,2] on a benchmark 25D-PestControl problem [3], with search space size 5^25. We provide this further analysis in Appendix E13 of the updated manuscript. This additional experiment clearly shows (see Figure 19) that GameOpt provides faster convergence to better solutions compared to these methods, supporting its integration with other high-dimensional methods and generalization to other combinatorial BO problem domains.

- **[Additional problem domain]:** We further experiment on the benchmark 25D-PestControl problem [3], with search space size 5^25. As given in Appendix E13 of the updated manuscript, we apply the GameOpt approach to this additional problem domain, which clearly shows (see Figure 19) that GameOpt generalizes well to other problem domains, providing an outperforming performance. This further supports GameOpt's applicability in various combinatorial BO settings.

- **[Batch size further ablations]:**  We also show the performance comparison under batch sizes $B=1$ and $B=3$ on Halogenase and GB1(4) domains, in Appendix-E7 of the updated manuscript. As shown in Figure-12, even under a more restricted setting on both problem domains, GameOpt variations achieve superior performance compared to the baselines.

Once again, we sincerely thank all reviewers for their thoughtful and constructive comments, which have helped us improve the quality of our work.

---

> [1] Changyong Oh, Jakub Tomczak, Efstratios Gavves, and Max Welling. Combinatorial bayesian optimization using the graph cartesian product. Advances in Neural Information Processing Systems, 32, 2019.

> [2] Leonard Papenmeier, Luigi Nardi, and Matthias Poloczek. Bounce: Reliable high-dimensional bayesian optimization for combinatorial and mixed spaces. Advances in Neural Information Processing Systems, 36:1764–1793, 2023.

> [3] Aryan Deshwal, Sebastian Ament, Maximilian Balandat, Eytan Bakshy, Janardhan Rao Doppa, and David Eriksson. Bayesian optimization over high-dimensional combinatorial spaces via dictionary-based embeddings. In International Conference on Artificial Intelligence and Statistics, pp. 7021–7039. PMLR, 2023.

---

### Meta-Review · Area_Chair_rqPj · 2024-12-21

**Metareview:**

The paper introduces GameOpt, a game-theoretical framework for combinatorial Bayesian optimization (CBO), with a focus on protein design. This work represents an innovative contribution by leveraging cooperative game theory to tackle scalability challenges in high-dimensional discrete optimization.

The main technical concerns from the reviewers revolves around the optimization gap between global/local optimum and Nash equilibrium, which I believe that the authors have addressed adequately. Some reviewers also noted the absence of comparisons with certain state-of-the-art methods, though the authors clarified the rationale behind their choice of baselines.

Overall, I believe this is a solid piece of work that could provide an alternative approach to the protein design community, and thus recommend acceptance.

**Additional Comments On Reviewer Discussion:**

NA

---

### Decision · Program_Chairs · 2025-01-22

Accept (Poster)